# The RNA-binding protein Puf5 and the HMGB protein Ixr1 contribute to cell cycle progression through the regulation of cell cycle-specific expression of *CLB1* in *Saccharomyces cerevisiae*

**Megumi Sato**[1,2]**, Kaoru Irie**[2]**, Yasuyuki Suda**[2,3]**, Tomoaki Mizuno**[2]**, Kenji Irie**[1,2]*

**1** Colledge of Medicine, School of Medicine and Health Sciences, University of Tsukuba, Tsukuba, Japan, **2** Department of Molecular Cell Biology, Graduate School of Comprehensive Human Sciences and Faculty of Medicine, University of Tsukuba, Tsukuba, Japan, **3** Live Cell Super-resolution Imaging Research Team, RIKEN Center for Advanced Photonics, Wako, Saitama, Japan

\* kirie@md.tsukuba.ac.jp

**Data Availability Statement:** Microarray data sets are available at the Gene Expression Omnibus at

## Abstract

Puf5, a Puf-family RNA-binding protein, binds to 3′ untranslated region of target mRNAs and negatively regulates their expression in *Saccharomyces cerevisiae*. The *puf5Δ* mutant shows pleiotropic phenotypes including a weakened cell wall, a temperature-sensitive growth, and a shorter lifespan. To further analyze a role of Puf5 in cell growth, we searched for a multicopy suppressor of the temperature-sensitive growth of the *puf5Δ* mutant in this study. We found that overexpression of *CLB2* encoding B-type cyclin suppressed the temperature-sensitive growth of the *puf5Δ* mutant. The *puf5Δ clb2Δ* double mutant displayed a severe growth defect, suggesting that Puf5 positively regulates the expression of a redundant factor with Clb2 in cell cycle progression. We found that expression of *CLB1* encoding a redundant B-type cyclin was decreased in the *puf5Δ* mutant, and that this decrease of the *CLB1* expression contributed to the growth defect of the *puf5Δ clb2Δ* double mutant. Since Puf5 is a negative regulator of the gene expression, we hypothesized that Puf5 negatively regulates the expression of a factor that represses *CLB1* expression. We found such a repressor, Ixr1, which is an HMGB (High Mobility Group box B) protein. Deletion of *IXR1* restored the decreased expression of *CLB1* caused by the *puf5Δ* mutation and suppressed the growth defect of the *puf5Δ clb2Δ* double mutant. The expression of *IXR1* was negatively regulated by Puf5 in an *IXR1* 3′ UTR-dependent manner. Our results suggest that *IXR1* mRNA is a physiologically important target of Puf5, and that Puf5 and Ixr1 contribute to the cell cycle progression through the regulation of the cell cycle-specific expression of *CLB1*.

## Author summary

Cell cycle progression is strictly regulated by cyclin-dependent kinases (CDKs). The activity of CDK is determined by the expression of cell cycle-specific cyclins. In this paper, we

http://www.ncbi.nlm.nih.gov/geo (GEO accession number GSE124908).

**Funding:** This research was supported by JSPS KAKENHI Grant Number 18K06053 and 22K06074 (to KI). The funders had no role in study design, data collection and analysis, decision to publish, or preparation of the manuscript.

**Competing interests:** The authors have declared that no competing interests exist.

demonstrate a novel regulatory mechanism for expression of a cyclin, *CLB1*, in budding yeast provoked by the RNA-binding protein Puf5 and the HMGB protein Ixr1. The RNA-binding protein Puf5 binds to the 3' untranslated region of *IXR1* mRNA, inducing sequential negative regulation of the Ixr1 protein levels. The Ixr1 protein negatively regulates *CLB1* expression via the *CLB1* promoter. This discovery of new regulators of cyclins indicates that the control of cyclin gene expression may be elaborately regulated in response to environmental conditions.

## Introduction

RNA-binding proteins play an important role in controlling the translation, turnover, and localization of mRNAs. RNA-binding proteins affect the fate of target mRNAs through binding to certain sequences found in the 5′ untranslated region (UTR), the coding region, or the 3′ UTR [1,2,3,4]. Among the three regions, the 3′ UTR has been reported to be the most common region where the sequences are found [4]. One well-known family of RNA-binding proteins is PUF (Pumilio and FBF) family. PUF family proteins are evolutionary conserved in many eukaryotes. In *Saccharomyces cerevisiae*, 6 Puf proteins have been reported [3,5]. Among them, we have focused on Puf5, also known as Mpt5. Puf5 binds to 3′ UTR of target mRNAs and recruits the Ccr4-Not complex to promote RNA degradation [6,7]. Puf5 has been reported to bind to more than 1,000 RNAs and regulate the expression of genes functioning in broad biological processes, including cell wall integrity (CWI) or cell cycle progression [8,9]. Among the binding targets, two physiologically important targets have been reported: one is *HO* mRNA encoding an endonuclease involved in mating-type switching [10], and the other is *LRG1* mRNA encoding a GTPase-activating protein (GAP) for Rho1 GTPase involved in regulating CWI [11–13]. Puf5 negatively regulates *LRG1* expression together with Ccr4 and controls the CWI pathway [11–15]. The *puf5Δ* mutant shows pleiotropic phenotypes including an abnormal mating-type switching in daughter cells [10], a weakened cell wall, a temperature-sensitive growth [11,14,15], and a shorter lifespan [5,14]. These pleiotropic effects indicate the multiple functions of Puf5 in cell growth, but the target mRNAs of Puf5 involved in growth control, other than *LRG1* mRNA, have not been fully understood.

Cell cycle progression is strictly regulated by a periodic change in the activity of cyclin-dependent kinase (Cdk). In *S. cerevisiae*, Cdc28 functions as a sole Cdk [16]. Activated Cdc28 phosphorylates various substrates necessary for the cell cycle progression, and this periodical regulation of the phosphorylation status of the proteins ensures a proper transition of the cell cycle. The main regulators of the periodicity of the Cdk activity are cyclin proteins. Cyclins are indispensable for Cdk activity, and the periodical change in the cyclin abundance controls the activity of Cdk [16]. In *S. cerevisiae*, there are two major groups of cyclins, G1 cyclins and B-type cyclins. G1 cyclins including Cln1, Cln2, and Cln3 promote the G1/S transition. B-type cyclins consist of six cyclins, Clb1, Clb2, Clb3, Clb4, Clb5, and Clb6. Clb5/Clb6 accumulate in the late G1 phase and contribute to initiating the S phase. Clb3/Clb4 are abundant in the S and G2 phases and promote spindle formation. The abundance of Clb1/Clb2 is increased in the G2 and M phases, and they play an important role in the transition from the G2 phase to the M phase. Clb2 performs as the major G2/M cyclin and is the most important of the six cyclin B proteins in mitosis [16]. On the other hand, the abundance of Clb2 is low in meiosis, and Clb1 acts as a major G2/M cyclin [17]. In regulating the cyclin abundance, both transcriptional control and protein degradation play crucial roles. B-type cyclin transcripts accumulate at the proper timing. As for G2/M cyclins, their transcription is regulated by cell cycle-dependent

activity of the Mcm1-Fkh2-Ndd1 activator complex [18,19]. Even though the importance of transcriptional activators has been reported, the contribution of transcriptional repressors to the regulation of the *CLB1/2* expression remains unclear.

In this study, we further analyzed the role of Puf5 in cell growth. We found that overexpression of *CLB2* suppressed the temperature-sensitive phenotype of the *puf5Δ* mutant, and that the *puf5Δ clb2Δ* double mutant showed a severe growth defect. In addition, the expression of *CLB1* encoding a redundant B-type cyclin with Clb2 was decreased in the *puf5Δ* mutant, and overexpression of *CLB1* recovered the growth defect of the *puf5Δ clb2Δ* mutant. We also found that deletion of *IXR1* encoding a transcriptional repressor suppressed the growth defect of the *puf5Δ clb2Δ* double mutant, and that the *ixr1Δ* mutation restored the reduced *CLB1* expression caused by the *puf5Δ* mutation. The expression of *IXR1* was negatively regulated by Puf5. Our data suggest that *IXR1* mRNA is a new physiologically important target of Puf5, and that the *IXR1* regulation engendered by Puf5 leads to the positive regulation of *CLB1* expression and assures the proper cell cycle progression.

## Materials and methods

### Strains and media

To manipulate DNA, the *Escherichia coli*, DH5α strain, was used. The *Saccharomyces cerevisiae*, W303 strain, was used as a yeast strain. The strains used in this study are described in S1 Table. Standard procedures were followed for yeast manipulations [20]. The medium used in this study includes YPD medium (2% Glucose, 2% bactopeptone, and 1% yeast extract) and synthetic complete medium (SC) [20]. For the strains needed to reduce osmolarity stress, 10% sorbitol was added. SC medium lacking amino acids (e.g., SC-Ura medium was SC medium lacking uracil) were used to select transformants.

### Plasmids

Plasmids used in this study are described in S2 Table. For the construction of YEplac195-*PUF5* plasmid, the fragment containing the *PUF5* gene together with upstream and downstream regions was amplified by PCR using genomic DNA as a template. The fragment was inserted between *Sal*I and *Eco*RI sites of YEplac195 plasmid. YEplac195-*CLB2* and YEplac195-*CLB1* plasmids were constructed in a similar manner. For the construction of YCplac33-*IXR1* plasmid, the fragment containing the *IXR1* gene together with upstream and downstream regions was amplified by PCR and inserted between *Sal*I and *Eco*RI sites of YCplac33 plasmid.

YCplac33-*CLB1*-HA was constructed as follows. The fragment containing *CLB1* promoter-*CLB1* ORF and the fragment containing *CLB1* 3´ UTR were amplified by PCR from genomic DNA, and the fragment encoding HA was amplified by PCR using the pFA6a-HA-kanMX6 plasmid as a template [21]. The recognition sequence of *Eco*RI was added to the top of the *CLB1* promoter-*CLB1* ORF fragment, and the sequence of *Sal*I was added to the end of the *CLB1* 3´ UTR fragment. To the other ends of the *CLB1* promoter-*CLB1* ORF fragment and the *CLB1* 3´ UTR fragment, the upper sequence and downstream sequence of HA fragment were added, respectively. Then, the fragments were inserted between *Sal*I and *Eco*RI sites of YCplac33 plasmid. YCplac33-*IXR1*-HA plasmid was constructed in a similar manner. YCplac33-*IXR1*-HA-*ADH1* 3´ UTR plasmid was constructed by inserting the fragment of *IXR1* promoter-*IXR1* ORF amplified from genomic DNA and the fragment of HA-*ADH1* 3´ UTR amplified from the pFA6a-HA-kanMX6 plasmid between *Sal*I and *Eco*RI sites of YCplac33 plasmid.

YCplac33-*CLB1* promoter -*GFP-ADH1* 3´ UTR and YCplac33-*CLB2* promoter -*GFP-ADH1* 3´ UTR were constructed as follows. The fragments corresponding to *CLB1*/

*CLB2* promoter were amplified by PCR from genomic DNA, and the *GFP-ADH1* 3′ UTR fragment was amplified by PCR from the pFA6a-GFP(S65T)-kanMX6 plasmid. Then, the fragments were inserted between *Eco*RI and *Sal*I site of YCplac33 plasmid. YCplac33-*MCM2* promoter -*GFP-IXR1* 3′ UTR were constructed using *MCM2* promoter, *GFP*, and *IXR1* 3′ UTR fragments amplified by PCR from genomic DNA, the pFA6a-GFP(S65T)-kanMX6 plasmid, and genomic DNA, respectively.

The pCgLEU2, pCgHIS3, and pCgTRP1 plasmids, which were pUC19 carrying the *Candida glabrata LEU2*, *HIS3*, and *TRP1* genes, respectively, were used to delete genes [22]. The pKl-*URA3* plasmid, pUC19 carrying the *Kluyveromyces lactis URA3* were also used for gene deletion.

## Gene deletion

Deletions of *PUF5*, *CLB2*, *CLB1*, *IXR1*, *BAR1*, and *LRG1* were constructed by PCR-based gene-deletion method [22–24]. Primer sets used in this study were listed in S3 Table. The fragments amplified by PCR were transformed into the wild-type strain and transformants were selected on the SC medium lacking the corresponding amino acids.

## Screening for multicopy suppressor of the temperature-sensitive phenotype of the *puf5Δ* mutant

The genomic DNA library in the YEp13 plasmid [25] was transformed into the *puf5Δ* mutant strain harboring pRS316-3xFLAG-LRG1 and plated on SC-Ura-Leu plates containing 10% sorbitol at room temperature for 4 days. pRS316-3xFLAG-LRG1 was transformed for the purpose of obtaining both downstream factors of Puf5 and other regulators of Lrg1. Then, the plates were replicated onto YPD plates and incubated at 37˚C for 3 days. Twenty-six transformants that formed colonies at 37˚C were identified. The corresponding plasmids were isolated from the transformants, and those that conferred the ability to proliferate at 37˚C to the *puf5Δ* strain were identified by retransformation. Four of twenty-six plasmids gave the ability to proliferate at 37˚C. Sequencing of the insert DNAs of the four recovered plasmids revealed that two contained the *SSD1* gene, one contained the *CLB2* gene, and one contained the *ZDS1* gene. Regional analysis of the suppression ability confirmed that the *SSD1*, *CLB2*, and *ZDS1* genes were responsible for ensuring the growth of the *puf5Δ* strain.

## Western blot analysis

Yeast cells were pre-cultured overnight in a liquid medium, then diluted to $OD_{600}$ = 0.5 (optical density measured at a wavelength of 600 nm), and cultured for 4 hours. Then, cells were collected by centrifuge and treated with sodium hydroxide for protein extraction [26]. Protein samples were loaded onto an SDS-PAGE gel for protein electrophoresis and then transferred to a PDVF membrane (Millipore). Then, the membrane was reacted with the primary antibody at 4˚C overnight and reacted with the secondary antibody for 1 hour at room temperature. The anti-HA monoclonal antibody HA11, the anti-Pgk1 monoclonal antibody 22C5D8 (Invitrogen), the anti-GFP monoclonal antibody (Roche), the anti-FLAG monoclonal antibody M2 (Sigma), and the anti-c-Myc monoclonal antibody 9E10 (Santa Cruz) were used as the primary antibodies. The anti-mouse IRDye 800CW secondary antibodies and IRDye 680RD secondary antibodies (LI-COR) were used as the secondary antibodies. The Pgk1 protein was used as a loading control. ODYSSEY CLx (LI-COR) was used to visualize and quantify the Clb1-HA protein, Ixr1-HA protein, GFP protein, Puf5-Myc protein, and Pgk1 protein. The fold change of the Clb1-HA, Ixr1-HA protein, GFP protein, and Puf5-Myc protein levels, normalized with the intensity of Pgk1, was calculated and statistically analyzed using Excel (Microsoft).

## RNA isolation, quantitative real-time PCR (qRT-PCR), and microarray analysis

Yeast cells were pre-cultured overnight in a liquid medium, then diluted to $OD_{600}$ = 0.5 (optical density measured at a wavelength of 600 nm), and cultured for 4 hours. Then, cells were collected by centrifuge. RNAs were extracted from the collected cells using ISOGEN reagent (Nippon Gene). From the extract, genomic DNAs were removed using RNeasy Mini kit (Qiagen), and reverse transcription was performed using the Prime Script RT reagent Kit (Takara). The cDNA levels were quantified by qRT-PCR using QuantStudio 5 (Thermo Fisher Scientific) with SYBR Premix Ex Taq (Takara). The primers used for the qRT-PCR were listed in S4 Table. The fold change of the mRNAs was calculated using *SCR1* or *ACT1* as internal control genes and statistically analyzed using Excel (Microsoft).

The microarray analysis was performed by the KURABO Bio-Medical Department using the Affymetrix GeneChip Yeast Genome 2.0 Array (Affymetrix, Santa Clara, California, USA). Microarray data sets are available at the Gene Expression Omnibus at http://www.ncbi.nlm. nih.gov/geo (GEO accession number GSE124908)

## Cell cycle synchronization by α-factor block and release

In the pheromone-induced cell cycle synchronization procedure, we used the *MAT***a** *bar1*Δ strains to prevent degradation of α-factor and followed the procedure previously reported [27]. Yeast cells were pre-cultured overnight in a YPD medium at 28˚C, then transferred into a fresh YPD medium, and cultured for 4 hours. When culturing strains harboring the YCplac33-*IXR1*-HA-*IXR1* 3´UTR plasmid, yeast cells were pre-cultured overnight in an SC-Ura medium at 28˚C, and then cells were collected by centrifuge, suspended in a small amount of YPD medium, and cultured at 28˚C for 2 hours. Sequentially, cells were transferred into a fresh YPD medium and cultured for 4 hours. We confirmed that more than 95% of cells carried the YCplac33-*IXR1*-HA-*IXR1* 3´ UTR plasmid in YPD culture. After the 4-hour culture, α-factor was added into the medium, and incubation was continued for 2 hours. Then, after collecting the 0-minute sample, cells were washed with a fresh YPD medium by centrifuge, transferred into a fresh YPD medium, and incubated at 28˚C. Samples were collected by centrifuge every 10 minutes from the time at release.

## RNA immunoprecipitation (RIP) analysis

To examine the interaction between Puf5 and *IXR1* mRNA, *ixr1*Δ *PUF5-3xFLAG-ADH1* 3´ UTR strains harboring YEplac195-*IXR1* with/without Puf5-binding element were used. Untagged *ixr1*Δ *PUF5* strains harboring the same plasmids were used as a negative control. Yeast cells were pre-cultured overnight in an SC-Ura medium, then diluted to $OD_{600}$ = 0.5 (optical density measured at a wavelength of 600 nm), and cultured for 4 hours. Then, cells were collected.

Cells were dissolved into the extraction buffer (0.3M XT buffer (0.05 M Hepes, 0.3 M KCl, and 0.004 M $MgCl_2$), protease inhibitors (APMSF and aprotinin), a reducing agent dithiothreitol, a detergent NP40, and RNasin (Promega)) and extracts were obtained by the beads crushing. The supernatant from the crude extracts was labeled as SUP samples. From the supernatant, Puf5-3xFLAG protein was immunoprecipitated with the anti-FLAG M2 affinity agarose gel (Sigma). After washing with extraction buffer with 10% glycerol, we obtained the immunoprecipitants labeled as beads samples. Protein and RNA were extracted from each sample, and the enrichment of *IXR1* mRNA was examined by qRT-PCR analysis.

### Statistical analysis

Graph data is presented as means ± standard error (SE) of biological samples. Statistical analyses were performed with Microsoft Office Excel using Tukey's test, and differences were regarded as significant when they were either $^{**}$P < 0.01 or $^{*}$P < 0.05.

## Results

### Overexpression of *CLB2* suppressed the temperature-sensitive phenotype of the *puf5Δ* mutant

Deletion of *PUF5* leads to a weakened cell wall, temperature-sensitive growth, and a short life span [11,14,15]. In order to further analyze the function of Puf5 in cell growth, we screened a genomic library for a gene whose overexpression suppresses the temperature-sensitive growth of the *puf5Δ* mutant. We identified *CLB2* encoding a B-type cyclin [16,17] and *ZDS1* involved in polarized growth [28] as a multicopy suppressor (for details of the screening, see Materials and methods). We also identified *SSD1*, the gene encoding a translational repressor contributing to polarized growth and CWI [14,29], in this screening, but we did not study the *SSD1* gene further here, since the W303 stains carry a defective *SSD1* allele. Overexpression of *PUF5* and *ZDS1* recovered the growth defect of the *puf5Δ* mutant at 35˚C and 37˚C (Fig 1A). However, multicopy *CLB2* suppressed the phenotype at 35˚C but not enough at 37˚C (Fig 1A). Therefore, *CLB2* seems to weakly suppress the phenotype of the *puf5Δ* mutant. Among three genes, we focused on the *CLB2*.

### The *puf5Δ clb2Δ* double mutant shows severe growth defect

Based on the result that overexpression of *CLB2* suppressed the temperature-sensitive phenotype of the *puf5Δ* mutant (Fig 1A), we considered two possibilities described in Fig 1B: the first is that *CLB2* functions downstream of Puf5 in the growth control, and the second is that *CLB2* has a redundant role with *PUF5* in the growth control. To clarify these possibilities, we genetically analyzed the interaction between *PUF5* and *CLB2*. We generated the *puf5Δ clb2Δ* double mutant and examined the cell growth. While the *puf5Δ* and the *clb2Δ* single mutants grew similarly to wild-type at 30˚C, the *puf5Δ clb2Δ* double mutant showed dramatical poor growth or lethality (Fig 1C). We also confirmed the severe growth defect of the *puf5Δ clb2Δ* double mutant by the overtime culture in a liquid medium. Consistently, the *puf5Δ clb2Δ* double mutant grew more slowly than wild-type, the *puf5Δ* mutant, or the *clb2Δ* mutant (Fig 1D).

The *clb2Δ* mutant has been reported to show an abnormally elongated shape and G2/M delay [30]. Regarding that, we examined whether the *puf5Δ* mutation accelerates this phenotype by observing the morphology of the *puf5Δ clb2Δ* double mutant. The *clb2Δ* mutant cells showed an elongated shape (Fig 1E, *clb2Δ*) as reported previously [30]. Corresponding to a previous report that the *puf5Δ* mutant shows a highly polarized shape [31], the *puf5Δ* mutant was enlarged compared to wild-type cells (Fig 1E, WT, *puf5Δ*). The *puf5Δ clb2Δ* double mutant was severely elongated (Fig 1E, *puf5Δ clb2Δ*), and the elongation was severer than that in the *clb2Δ* mutant. Nuclear staining patterns of the *puf5Δ clb2Δ* double mutant showed partial defects in chromosome segregation (Fig 1E, *puf5Δ clb2Δ* DAPI). Taken together, these results suggest that Clb2 functions in parallel with Puf5 in cell growth rather than acts as a downstream factor of Puf5.

### *CLB1* expression was diminished in the *puf5Δ* mutant

Since Puf5 is an RNA-binding protein that binds to 3′ UTR of target mRNA [3,5], it is unlikely that Puf5 directly functions as a redundant factor with Clb2. Rather, it is more likely that Puf5

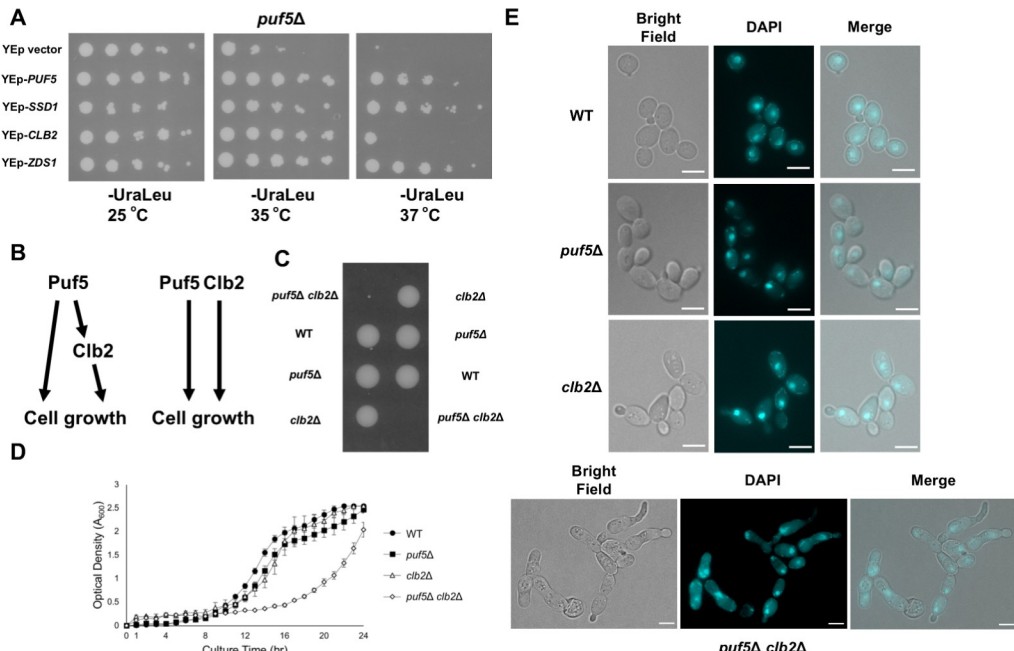

**Fig 1. Overexpression of *CLB2* repressed the temperature-sensitive phenotype of the *puf5Δ* mutant.** (A) The multi-copy suppressors of the temperature-sensitive phenotype of the *puf5Δ* mutant. The *puf5Δ* mutants harboring YEp13, YEp13-*PUF5*, YEp13-*SSD1*, YEp13-*CLB2*, or YEp13-*ZDS1* in addition to the plasmids pRS316-3xFLAG-*LRG1* were cultured at 25˚C, serially diluted, and spotted onto an SC-Ura-Leu plate, and incubated for 4 days at 25˚C, 35˚C, or 37˚C. (B) Models showing how Puf5 and Clb2 function in cell growth. One is that Clb2 is a downstream target of Puf5, and the other is that Puf5 and Clb2 function in a parallel manner. (C) The tetrad analysis of the strains that are heterozygous for the alleles of *PUF5* and *CLB2*. The cells were sporulated, dissected on a YPD plate, and cultured at 30˚C for 3 days. (D) The growth curve of wild-type (black circle), the *puf5Δ* mutant (black square), the *clb2Δ* mutant (white triangle), and the *puf5Δ clb2Δ* mutant (white rhombus) at 25˚C. The strains were pre-cultured in a YPD medium containing 10% sorbitol overnight at 25˚C, and then transferred into a fresh YPD-Sorbitol medium and cultured at 25˚C for 1 day. The data shows the mean± SE (n = 3) of the optical density. (E) Morphology of wild-type, the *puf5Δ* mutant, the *clb2Δ* mutant, and the *puf5Δ clb2Δ* double mutant cells. Bright-field (left), DAPI staining (middle), and the overlayed (right) were shown. The scale bar represents 5 μm.

regulates expression of a target gene that has a redundant role with Clb2 in cell cycle progression, and that expression of the target gene is decreased in the *puf5Δ* mutant. In order to find a target, we conducted microarray analysis using wild-type and the *puf5Δ* mutant and screened the genes that exhibited a certain decrease in expression (log2 fold change ≦ -0.5) upon *PUF5* deletion. In the *puf5Δ* mutant, mRNA levels of 59 genes were decreased (log2 fold change ≦ -0.5). Among 59 genes, expression of *CLB1* and *CLB6* encoding different B-type cyclins were decreased to approximately 50% and 60%, respectively (Table 1). Expression of other cyclin genes, *CLB2*, *CLB3*, *CLB4*, *CLB5*, *CLN1*, *CLN2*, and *CLN3* did not show a significant change in the *puf5Δ* mutant (Table 1). The whole data of this microarray analysis is available in Valderrama et al. [32].

To confirm the microarray results, we examined the *CLB1* and *CLB6* mRNA levels in wild-type and the *puf5Δ* mutant by qRT-PCR. The *CLB1* mRNA level in the *puf5Δ* mutant was decreased to 52% of that in wild-type with a statistically significant difference (Fig 2A). The *CLB6* mRNA levels were not statistically significantly different between the *puf5Δ* mutant and wild-type (Fig 2B). Therefore, we further analyzed the regulation of the *CLB1* expression by Puf5. While deletion of *PUF5* caused the decrease in the *CLB1* mRNA level, overexpression of *PUF5* in wild-type cells increased the *CLB1* mRNA level (Fig 2C). We also examined Clb1 protein levels in wild-type and the *puf5Δ* mutant. Clb1-HA protein level in the *puf5Δ* mutant was

Table 1. **The extract data of the expression of *CLB* and *CLN* from previous microarray data.** The microarray data of each gene in wild-type and the *puf5Δ* mutant strains are presented. The whole data is presented in (Valderrama et al., 2021) [32]. The log2 fold change of the expression of each gene in the *puf5Δ* mutant normalized to that in wild-type are shown within parenthesis.

| GENE | wild-type | *puf5Δ* |
|---|---|---|
| *CLB1* | 694.6 (1) | 411.8 (-0.75) |
| *CLB2* | 569.4 (1) | 585.5 (0.04) |
| *CLB3* | 637.7 (1) | 507.6 (-0.33) |
| *CLB4* | 347.7 (1) | 291.1 (-0.26) |
| *CLB5* | 557.7 (1) | 502.4 (-0.15) |
| *CLB6* | 270.5 (1) | 172.1 (-0.65) |
| *CLN1* | 701.1 (1) | 661 (-0.08) |
| *CLN2* | 620.9 (1) | 667.7 (0.10) |
| *CLN3* | 1251.1 (1) | 1096.5 (-0.19) |

statistically significantly decreased to 24% of that in wild-type (Fig 2D and 2E). These results suggest that Puf5 positively regulates *CLB1* expression.

## The decrease of the *CLB1* expression in the *puf5Δ* mutant contributes to the growth defect of the *puf5Δ clb2Δ* double mutant

Above data suggested that Puf5 positively regulates *CLB1* expression, a paralog of *CLB2*, encoding a G2/M cyclin (Fig 3A). To confirm that *CLB1* acts downstream of Puf5, we deleted *PUF5* and *CLB1* in a diploid cell and performed tetrad analysis. Each single mutant, the *puf5Δ* mutant or the *clb1Δ* mutant, grew as well as wild-type (Fig 3B *puf5Δ*, *clb1Δ*, WT). The growth of the *puf5Δ clb1Δ* double mutant did not apparently differ from that of single mutants. These data corroborate the model that *CLB1* is a downstream factor of Puf5. Regarding that Puf5 positively regulates *CLB1* expression, we asked if this control influences cell growth together with Clb2 (Fig 3A). To make it clear, we exogenously overexpressed *CLB1* and *CLB2* in the *puf5Δ clb2Δ* double mutant. Overexpression of either *CLB1* or *CLB2* suppressed the growth defect of the *puf5Δ clb2Δ* double mutant, although the growth recovery by *CLB1* overexpression was weaker than that by *CLB2* overexpression (Fig 3C). We also examined whether overexpression of *CLB1* or *CLB2* suppresses the elongated phenotype of the *puf5Δ clb2Δ* double mutant. Consistent with the observation shown in Fig 1E, the *puf5Δ clb2Δ* double mutant harboring YEp vector showed severely elongated morphology (Fig 3D, *puf5Δ clb2Δ* [YEp vector]). When *CLB1* or *CLB2* was overexpressed, the elongated phenotype was suppressed (Fig 3D, *puf5Δ clb2Δ* [YEp-*CLB1*], *puf5Δ clb2Δ* [YEp-*CLB2*]). However, the cell sizes were still larger in the *CLB1*-overexpressed strain than in the *CLB2*-overexpressed one. This may be reflected in the partial restoration of the growth defect of the *puf5Δ clb2Δ* double mutant by overexpression of *CLB1* (Fig 3C). These results supported the model that the positive regulation of *CLB1* expression by Puf5 plays a key role in cell growth when Clb2 is absent. To further confirm this model, we deleted *PUF5*, *CLB2*, *CLB1*, and *LRG1* in diploid cell and performed tetrad analysis. Since we and other groups have previously reported that *LRG1* deletion suppresses the high

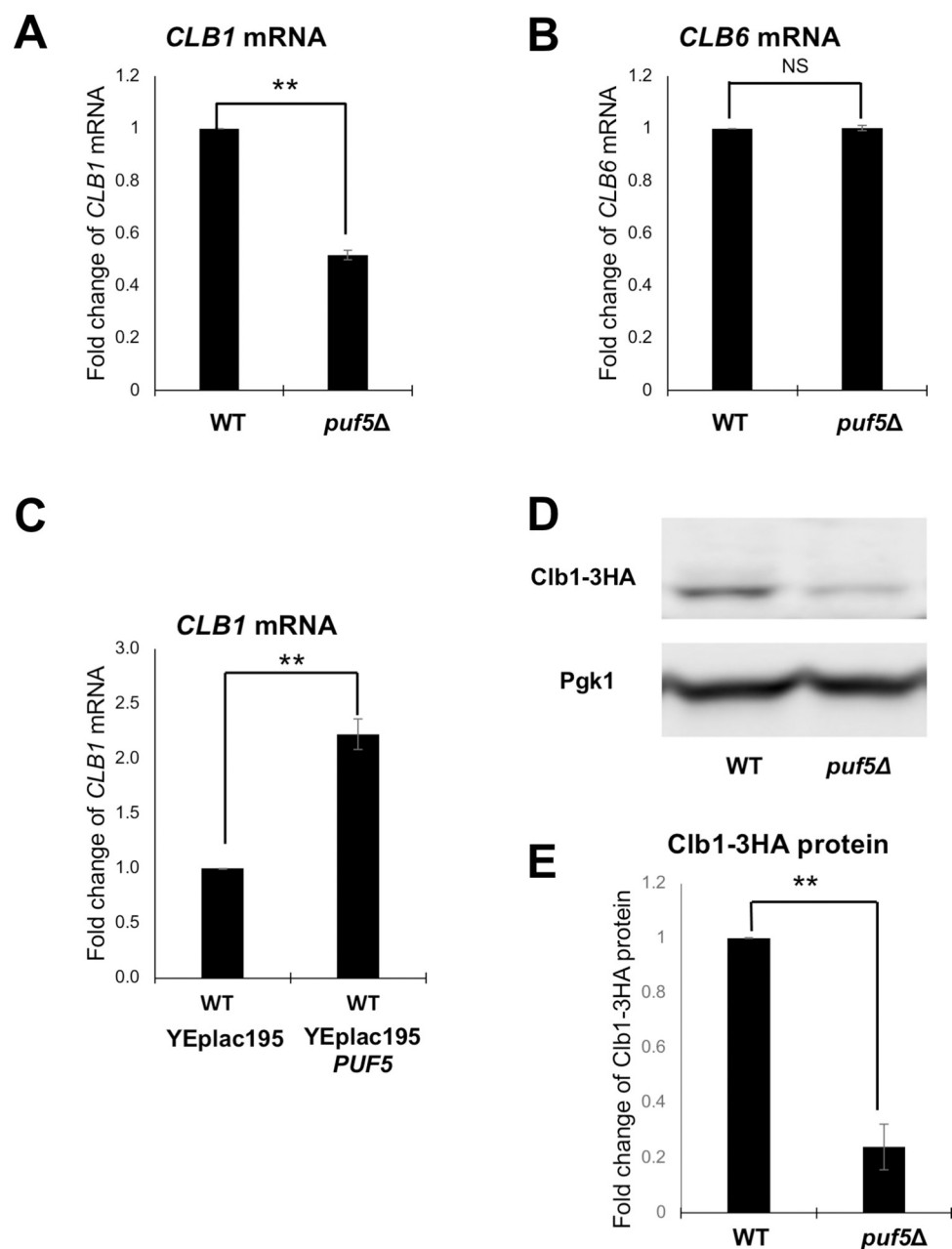

**Fig 2. Puf5 positively regulates *CLB1* expression.** (A, B) The mRNA levels of *CLB1* (A) and *CLB6* (B) in wild-type and the *puf5*Δ mutant. The cells were cultured in a YPD medium containing 10% sorbitol at 25˚C until the log phase. The *CLB* mRNA levels were quantified by qRT-PCR analysis, and the relative mRNA levels were calculated using the *SCR1* reference gene. The data shows the mean ± SE (n = 3) of the fold change of *CLB1* mRNA (A) and *CLB6* mRNA (B) relative to the mRNA level in wild-type. *P < 0.05, **P < 0.01 as determined by Tukey's test. (C) The mRNA levels of *CLB1* in the wild-type cell overexpressing *PUF5*. The wild-type strains harboring plasmids YEplac195 or YEplac195-*PUF5* were cultured in an SC-Ura medium at 28˚C until the exponential phase. The *CLB1* mRNA levels were quantified by qRT-PCR analysis, and the relative mRNA levels were calculated using the *SCR1* reference gene. The data shows the mean ± SE (n = 3) of the fold change of *CLB1* mRNA relative to the mRNA level in wild-type harboring the YEplac195 plasmid. *P < 0.05, **P < 0.01 as determined by Tukey's test. (D, E) The Clb1 protein levels in wild-type and the *puf5*Δ mutant, and the quantitative analysis data of Clb1-HA protein level. The strains harboring the YCplac33-*CLB1-HA-CLB1* 3´ UTR plasmid were cultured in an SC-Ura medium at 28˚C. The extracts were immunoblotted with anti-HA antibody or anti-Pgk1 antibody. Clb1-HA protein level was quantified and normalized with the Pgk1 protein level. The data shows the mean ± SE (n = 3) of the fold change of Clb1-HA protein relative to the protein level in wild-type (E). *P < 0.05, **P < 0.01 as determined by Tukey's test.

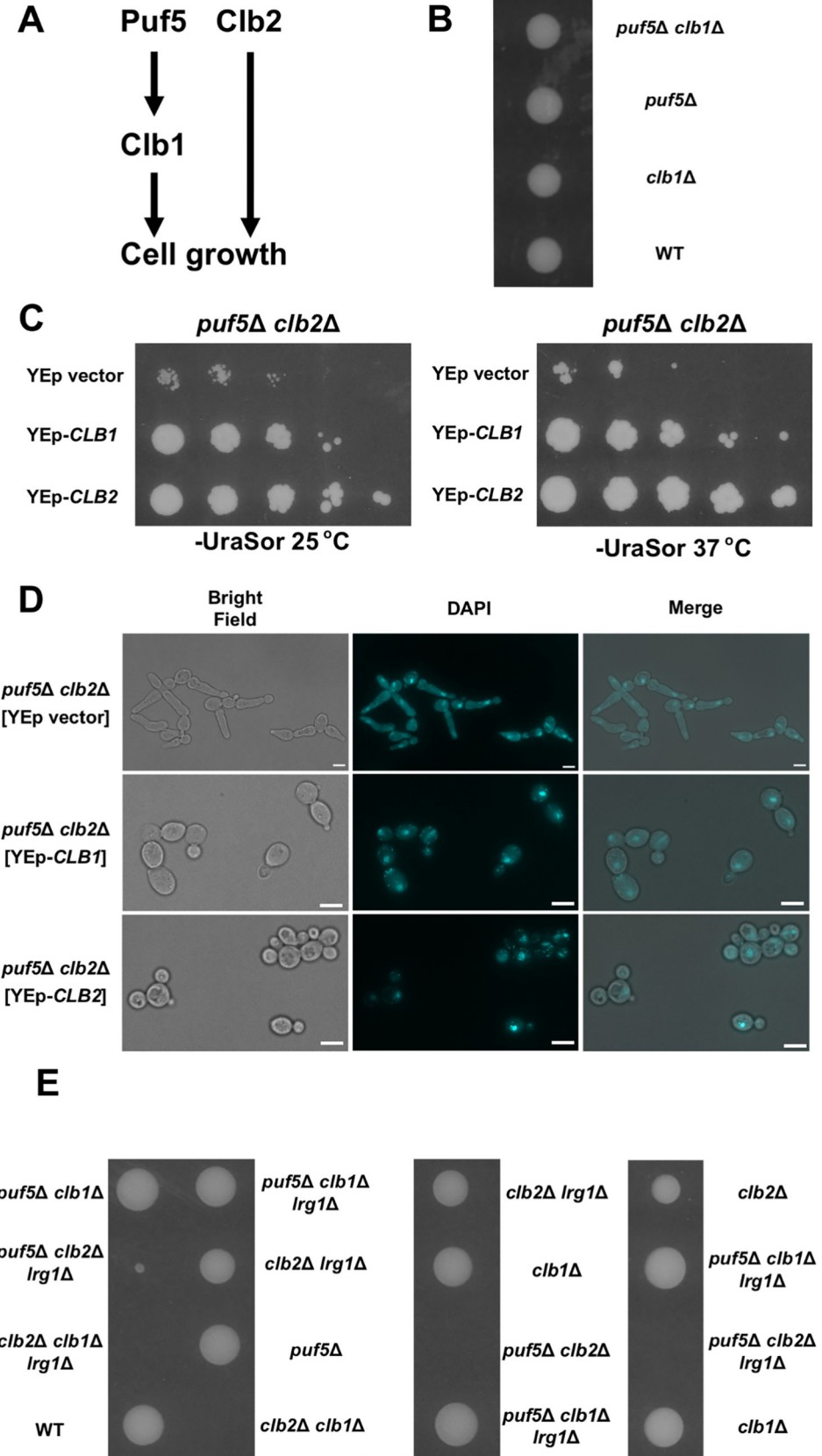

**Fig 3. The decrease of *CLB1* expression in the *puf5Δ* mutant contributes to the growth defect of the *puf5Δ clb2Δ* double mutant.** (A) A model of the Puf5 function in cell growth showing the hypothesis that Puf5 controls cell growth through the positive regulation of *CLB1*, a paralog of *CLB2*. (B) The tetrad analysis of the strains that are heterozygous

for the alleles of *PUF5* and *CLB1*. The cells were sporulated, dissected on a YPD plate, and cultured at 30˚C for 3 days. (C) The effects of overexpression of *CLB1* and *CLB2* are presented. The *puf5Δ clb2Δ* double mutant strains harboring plasmids YEplac195, YEplac195-*CLB1*, or YEplac195-*CLB2* were cultured in an SC-Ura medium containing 10% sorbitol at 25˚C until the exponential phase. Cells were serially diluted, spotted onto an SC-Ura plate containing 10% sorbitol, and incubated for 4 days at 25˚C or 37˚C. (D) Morphology of the *puf5Δ clb2Δ* double mutant strains harboring YEp vector, YEp195-*CLB1*, or YEp195-*CLB2*. Bright-field (left), DAPI staining (middle), and overlayed (right) were shown. The scale bar represents 5μm. (E) The tetrad analysis of the strains that are heterozygous for the alleles of *PUF5*, *CLB2*, *CLB1*, and *LRG1*. The cells were sporulated, dissected on a YPD plate, and cultured at 30˚C for 3 days.

temperature-sensitive growth of the *puf5Δ* mutant [11,14,15], we considered a possibility that Lrg1 is involved in the growth defect of the *puf5Δ clb2Δ* double mutant. Consistent with the results shown in Fig 3B, the *clb1Δ* single and the *puf5Δ clb1Δ* double mutant grew as well as wild-type (Fig 3E *clb1Δ*, *puf5Δ clb1Δ*, WT). The *clb2Δ* single mutant showed slower growth than wild-type, but the difference was not remarkable (Fig 3E *clb2Δ*, WT). As also shown in Fig 1C, the *puf5Δ clb2Δ* double mutant showed a severe growth defect or lethality (Fig 3E *puf5Δ clb2Δ*). The *clb2Δ clb1Δ* double mutant was also lethal (Fig 3E *clb2Δ clb1Δ*) as previously reported [33]. Deletion of *LRG1*, another target of Puf5, did not affect the phenotype of the *puf5Δ clb2Δ* double mutant and the *clb2Δ clb1Δ* double mutant (Fig 3E *puf5Δ clb2Δ lrg1Δ*, *clb2Δ clb1Δ lrg1Δ*). These results suggest that each of Clb2 and Clb1 complements their function each other to restore the decrease in the amount of G2/M cyclin. Deletion of *PUF5* showed a severe growth defect only in combination with *CLB2* deletion and had no effect in combination with *CLB1* deletion, and this growth defect was independent of *LRG1*, another target of Puf5. Thus, the positive regulation of *CLB1* expression by Puf5 was physiologically important when *CLB2* expression was lost.

## Puf5 regulates *CLB1* expression through the *CLB1* promoter

Since Puf5 regulates *CLB1* expression positively rather than negatively, we investigated whether Puf5 regulates *CLB1* expression at the transcriptional level. For this purpose, we constructed a reporter plasmid, the *CLB1* promoter-*GFP-ADH1* 3′ UTR plasmid that harbors the *GFP* driven by the *CLB1* promoter. The *ADH1* 3′ UTR was used in this reporter plasmid, since the expression of *ADH1* was not affected by *PUF5* deletion (S1A Fig). We examined the expression of *GFP* mRNA driven by the *CLB1* promoter in wild-type and the *puf5Δ* mutant by qRT-PCR. As a result, the *GFP* mRNA level in the *puf5Δ* mutant was decreased to 46% of that in wild-type with a statistically significant difference (Fig 4A). This data corresponds to the decrease of endogenous *CLB1* mRNA level in the *puf5Δ* mutant (Fig 2A). To confirm that the decrease of *GFP* mRNA level in the *puf5Δ* mutant is dependent on the *CLB1* promoter, we also constructed another reporter plasmid, the *CLB2* promoter-*GFP-ADH1* 3′ UTR plasmid, and examined the *GFP* mRNA levels in wild-type and the *puf5Δ* mutant. We used the *CLB2* promoter in consideration of the data that *CLB2* mRNA is expressed to the same extent between wild-type and the *puf5Δ* mutant (S1B Fig). The expression of *GFP* mRNA driven by the *CLB2* promoter did not show a statistically significant difference between wild-type and the *puf5Δ* mutant (Fig 4B). These data suggest that Puf5 positively regulates *CLB1* expression via the *CLB1* promoter at the transcriptional level.

## The HMGB protein Ixr1 functions downstream of Puf5 and contributes to the growth defect of the *puf5Δ clb2Δ* mutant

For the mechanism of how Puf5 positively regulates *CLB1* transcription, we hypothesized that Puf5 negatively regulates expression of a transcriptional repressor of *CLB1* (Fig 5A), regarding that Puf5 is thought to be a negative regulator of gene expression [10–12,15].

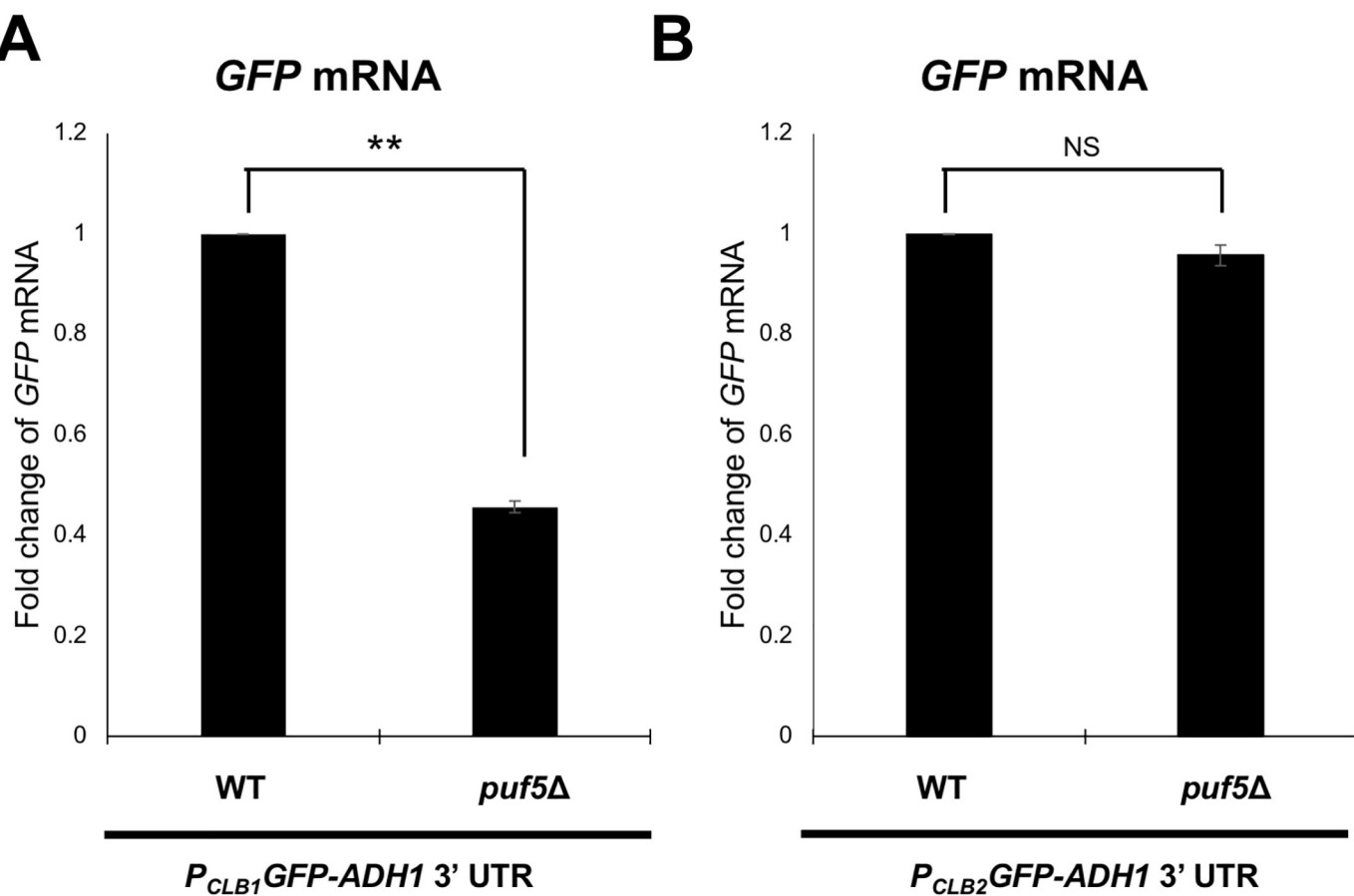

**Fig 4. Puf5 positively regulates *CLB1* transcription through the *CLB1* promoter.** (A, B) The mRNA levels of *GFP* in wild-type and the *puf5*Δ mutant. The strains harboring the YCplac33-*CLB1* promoter-*GFP-ADH1* 3′ UTR plasmid (A) or YCplac33-*CLB2* promoter-*GFP-ADH1* 3′ UTR plasmid (B) were cultured in an SC-Ura medium at 28°C until the exponential phase. The *GFP* mRNA levels were quantified by qRT-PCR analysis, and the relative mRNA levels were calculated using the *SCR1* reference gene. The data shows the mean ± SE (n = 3) of the fold change of *GFP* mRNA relative to the mRNA level in wild-type. *P < 0.05, **P < 0.01 as determined by Tukey's test.

We searched the *Saccharomyces* Genome Database (SGD https://www.yeastgenome.org/) and found 16 genes encoding a regulator of *CLB1*. We examined the mRNA levels of these 16 genes in wild-type and the *puf5*Δ mutant by qRT-PCR, and found that the expression of *IXR1*, *FKH1*, *FKH2*, *HFI1*, *HIR1*, and *STE12* genes was increased (fold change > 1.5) in the *puf5*Δ compared to wild-type (S5 Table). *IXR1* encodes an HMGB (High Mobility Group box B) protein. [34,35]. Fkh1/ Fkh2 are members of the winged-helix/forkhead (FOX) transcription factor gene family and regulate the transcription of *CLB1* and *CLB2* in the G2/M phase [18,36]. Hfi1 is a component of the SAGA complex [37], Hir1 is a corepressor involved in the transcriptional regulation of histone gene [38], and Ste12 is a MAPK cascade-activated transcription factor taking a part in the pheromone response and pseudohyphal growth [39]. We deleted each gene in addition to *PUF5*, *CLB2*, and *LRG1* genes in diploid cells and performed tetrad analysis to examine whether the deletion recovered the severe growth defect of the *puf5*Δ *clb2*Δ double mutant. *LRG1* was additionally deleted in this tetrad analysis for the purpose of ensuring that the regulator of *CLB1* expression acts independently of *LRG1*. Consequently, we found that deletion of *IXR1* recovered the growth defect of *puf5*Δ *clb2*Δ double mutant (Fig 5B *puf5*Δ *clb2*Δ, *puf5*Δ *clb2*Δ *ixr1*Δ). This recovery was also observed in the *lrg1*Δ background (Fig 5B *puf5*Δ *clb2*Δ *ixr1*Δ, *puf5*Δ *clb2*Δ *lrg1*Δ *ixr1*Δ). None of the deletions of

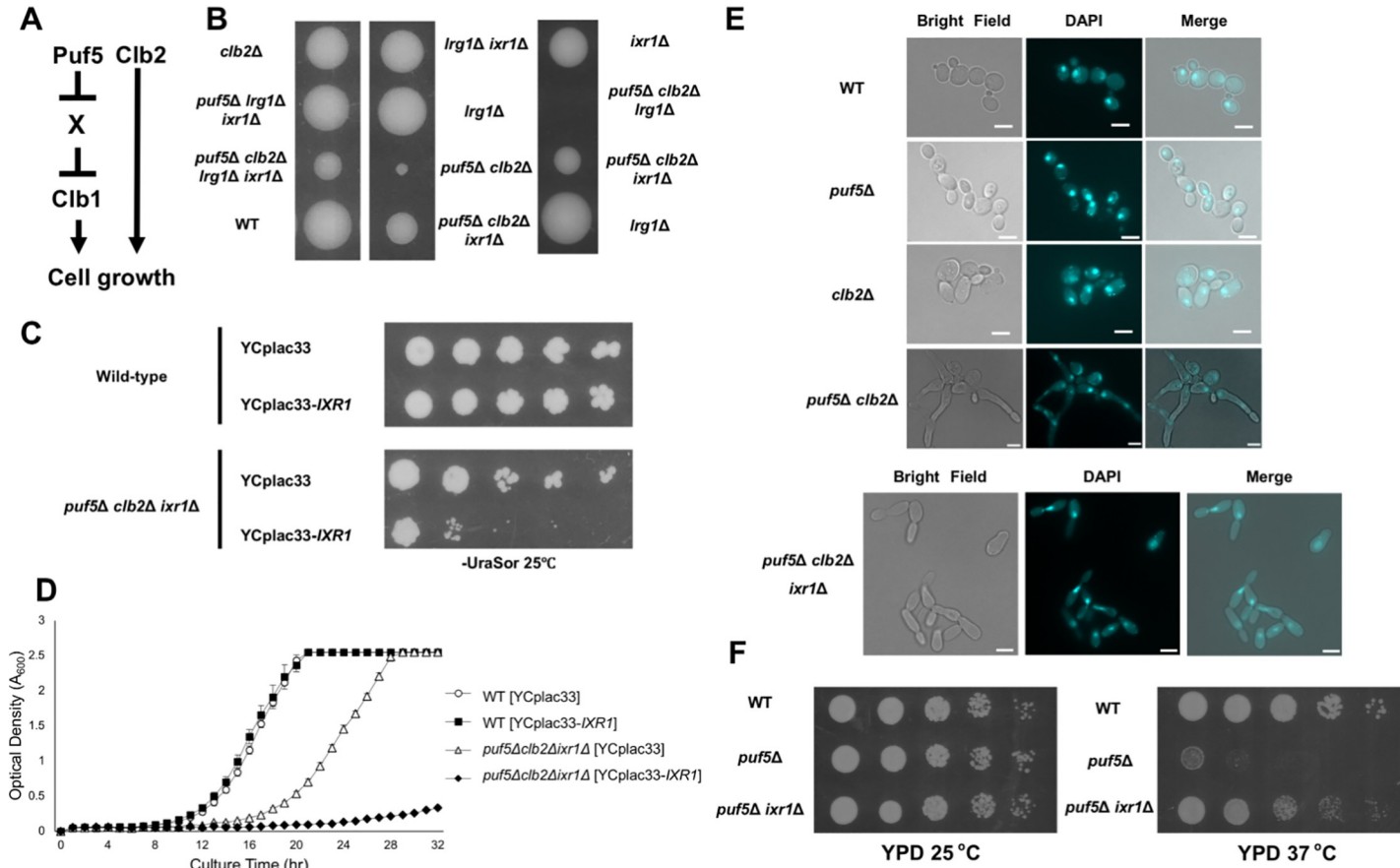

**Fig 5. Deletion of *IXR1* recovered the growth defect of the *puf5Δclb2Δ* double mutant.** (A) A model showing how Puf5 functions in cell growth through the *CLB1* regulation. The hypothesis is presented that Puf5 negatively regulates the expression of X, and X negatively regulates *CLB1* expression. (B) The tetrad analysis of the strains that are heterozygous for the alleles of *PUF5*, *CLB2*, *IXR1*, and *LRG1*. The cells were sporulated, dissected on a YPD plate, and cultured at 30°C for 3 days. (C) The spot assay of wild-type and the *puf5Δ clb2Δ ixr1Δ* triple mutant harboring plasmids YCplac33 or YCplac33-*IXR1*. The strains were cultured in an SC-Ura medium containing 10% sorbitol at 25°C until the exponential phase and collected. Cells were serially diluted, spotted onto an SC-Ura plate containing 10% sorbitol, and incubated for 4 days at 25°C. (D) The growth curve of wild-type and the *puf5Δ clb2Δ ixr1Δ* triple mutant harboring plasmids YCplac33 or YCplac33-*IXR1* at 25°C. The strains were pre-cultured in an SC-Ura medium overnight at 25°C, then transferred into a fresh SC-Ura medium, and cultured at 25°C for 1.5 days. The data shows the mean± SE (n = 3) of the optical density. The white circle markers show wild-type [YCplac33], the black square markers show wild-type [YCplac33-*IXR1*], the white triangle markers show the *puf5Δ clb2Δ ixr1Δ* [YCplac33], and the black rhombus markers show the *puf5Δ clb2Δ ixr1Δ* [YCplac33-*IXR1*]. (E) Morphology of wild-type, the *puf5Δ* mutant, the *clb2Δ* mutant, the *puf5Δ clb2Δ* double mutant, and the *puf5Δ clb2Δ ixr1Δ* triple mutant strains. Bright field (left), DAPI staining (middle), and overlayed (right) were shown. The scale bar represents 5 μm. (F) The spot assay of wild-type, the *puf5Δ* mutant, and the *puf5Δ ixr1Δ* double mutant. The strains were cultured in a YPD medium at 25°C until the exponential phase. Cells were serially diluted, spotted onto YPD plate and incubated for 1 day at 25°C or 37°C.

other regulatory genes recovered the growth defect of the *puf5Δ clb2Δ* double mutant (S2A–S2E Fig). Fkh1 and Fkh2 have been reported to regulate the expression of *CLB1* and *CLB2* [18,36], but loss of *FKH1* or *FKH2* also did not impair the growth of *puf5Δ* mutant or *puf5Δ clb2Δ* double mutant (S2A and S2B Fig). Therefore, we focused on the *IXR1* gene. *IXR1* encodes a transcriptional repressor with the HMG domain and has been found as a factor binding to cisplatin-DNA adducts [34,35,40,41]. Ixr1 represses the expression of hypoxic genes during normoxia and regulates hypoxic response [42]. While the *ixr1Δ* mutation restored the growth defect of the *puf5Δ clb2Δ* double mutant, this restoration was not observed at 37°C (S3 Fig *puf5Δ clb2Δ*, *puf5Δ clb2Δ ixr1Δ*). Since the *lrg1Δ* mutation retrieved the growth defect of the *puf5Δ clb2Δ ixr1Δ* triple mutant (S3 Fig *puf5Δ clb2Δ ixr1Δ*, *puf5Δ clb2Δ lrg1Δ ixr1Δ*), the *ixr1Δ* mutation and the *lrg1Δ* mutation seem to affect cell growth independently.

To confirm the influence of *IXR1* on the growth defect of the *puf5Δ clb2Δ* double mutant, we introduced YCplac33-*IXR1* in the *puf5Δ clb2Δ ixr1Δ* triple mutant. The expression of *IXR1* suppressed the growth of the *puf5Δ clb2Δ ixr1Δ* triple mutant (Fig 5C). This result was confirmed by the overtime culture in a liquid medium (Fig 5D). We also examined whether *IXR1* deletion restored not only the growth defect of the *puf5Δ clb2Δ* double mutant but also its elongated morphology. Consistent to the results shown in Fig 1E, the *clb2Δ* mutant cells showed an elongated shape compared to wild-type cells, and the *puf5Δ* mutation accelerated the elongation (Fig 5E WT, *clb2Δ*, *puf5Δ clb2Δ*). The *puf5Δ clb2Δ ixr1Δ* triple mutant cells were still elongated, but the degree of the elongation was much smaller than in the *puf5Δ clb2Δ* double mutant (Fig 5E *puf5Δ clb2Δ ixr1Δ*, *puf5Δ clb2Δ*). These results suggest that Ixr1 functions downstream of Puf5 and contributes to the growth defect of the *puf5Δ clb2Δ* double mutant.

To further investigate whether Ixr1 acts downstream of Puf5, not Clb2, we examined whether the *ixr1Δ* mutation suppresses the temperature-sensitive phenotype of the *puf5Δ CLB2⁺* mutant. The *puf5Δ* mutant grew poorly at 37˚C compared to wild-type (Fig 5F). Although the effect was not enough to completely suppress the temperature-sensitiveness, the *ixr1Δ* mutation recovered the growth defect of the *puf5Δ* mutant at 37˚C (Fig 5F). This data supports the model that Ixr1 functions downstream of Puf5.

## Ixr1 negatively regulates the *CLB1* transcription

We next examined whether Ixr1 negatively regulates *CLB1* expression (Fig 6A). First, we examined whether the *ixr1Δ* mutation restored the decreased expression of *CLB1* in the *puf5Δ clb2Δ* double mutant. The *CLB1* mRNA levels was decreased in the *puf5Δ* mutant and the *puf5Δ clb2Δ* double mutant with a statistically significant difference, and this decreased expression of *CLB1* mRNA was restored in the *puf5Δ clb2Δ ixr1Δ* triple mutant (Fig 6B). When we examined the *CLB1* mRNA levels in the *puf5Δ clb2Δ ixr1Δ* triple mutant harboring plasmids YCp vector, YCp-*IXR1*, or YEp-*IXR1*, the *CLB1* mRNA levels in the *puf5Δ clb2Δ ixr1Δ* triple mutant harboring YCp-*IXR1* and YEp-*IXR1* was significantly decreased to 53% and 37%, respectively, of that in the triple mutant with YCp vector (Fig 6C). These results suggest that *IXR1* deletion restored the decreased expression of *CLB1* caused by the *puf5Δ* mutation and simultaneously recovered the severe growth defect of the *puf5Δ clb2Δ* double mutant as shown in Fig 5B–5E.

Next, we examined the endogenous *CLB1* mRNA level in the *ixr1Δ* mutant. Consistent with the results in Fig 2A, the *CLB1* mRNA level in the *puf5Δ* mutant was decreased to 53% of that in wild-type (Fig 6D). In the *ixr1Δ* mutant, the *CLB1* mRNA level was approximately 1.4-times as much as that in wild-type (Fig 6D). We also investigated the expression of the reporter plasmid, *CLB1* promoter-*GFP-ADH1* 3′ UTR, in the *ixr1Δ* mutant to clarify whether Ixr1 negatively regulates *CLB1* expression through the *CLB1* promoter. Since the expression of *ADH1* mRNA was not influenced by deletion of *PUF5* or *IXR1* (S1A Fig), we utilized the *ADH1* 3' UTR. The expression of *GFP* mRNA driven by the *CLB1* promoter was 1.8-times increased in the *ixr1Δ* mutant than that in wild-type with a statistically significant difference (Fig 6E). To confirm that Ixr1 specifically regulates the *CLB1* promoter, we also examined the *GFP* mRNA level driven by the *CLB2* promoter in wild-type, the *puf5Δ* mutant, the *ixr1Δ* mutant, and the *puf5Δ ixr1Δ* double mutant, since the *CLB2* mRNA level was not affected by *IXR1* deletion (S1B Fig). As a result, the expression of *GFP* mRNA driven by the *CLB2* promoter did not show a statistically significant difference among the four strains examined (Fig 6F). These results suggest that Ixr1 negatively regulates *CLB1* transcription. When we compare the expression of both endogenous *CLB1* and *GFP* reporter between the *ixr1Δ* mutant and the

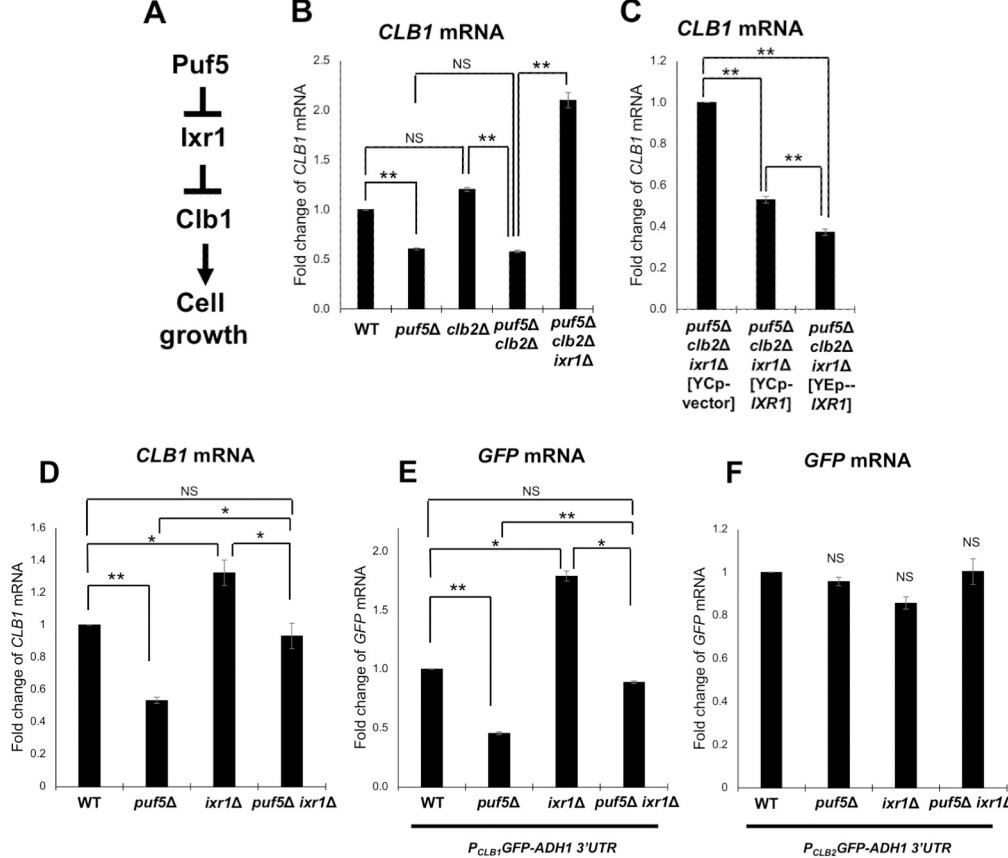

**Fig 6. Ixr1 is a downstream factor of Puf5 and negatively regulates *CLB1* expression.** (A) A model showing how Puf5 functions in cell growth through the *CLB1* regulation mediated by Ixr1. The hypothesis is presented that Puf5 negatively regulates the expression of the Ixr1 repressor, and Ixr1negatively regulates the *CLB1* transcription. (B)The *CLB1* mRNA levels in wild-type, the *puf5Δ* mutant, the *clb2Δ* mutant, the *puf5Δ clb2Δ* double mutant, and the *puf5Δ clb2Δ ixr1Δ* triple mutant. The cells were cultured in a YPD medium containing 10% sorbitol at 28˚C until the exponential phase. The *CLB1* mRNA levels were quantified by qRT-PCR analysis, and the relative mRNA levels were calculated using the *SCR1* reference gene. The data shows the mean ± SE (n = 5) of the fold change of *CLB1* mRNA relative to the mRNA level in wild-type. *P < 0.05, **P < 0.01 as determined by Tukey's test. (C) The *CLB1* mRNA levels in the *puf5Δ clb2Δ ixr1Δ* mutant harboring plasmids YCplac33, YCplac33-*IXR1*, or YEplac195-*IXR1*. The strains were cultured in an SC-Ura medium at 25˚C until the exponential phase. The *CLB1* mRNA levels were quantified by qRT-PCR analysis, and the relative mRNA levels were calculated using the *SCR1* reference gene. The data shows the mean ± SE (n = 5) of the fold change of *CLB1* mRNA relative to the mRNA level in the strain harboring the YCplac33 plasmid. *P < 0.05, **P < 0.01 as determined by Tukey's test. (D) The *CLB1* mRNA levels in wild-type, the *puf5Δ* mutant, the *ixr1Δ* mutant, and the *puf5Δ ixr1Δ* double mutant. The cells were cultured in a YPD medium at 28˚C until the exponential phase. The *CLB1* mRNA levels were quantified by qRT-PCR analysis, and the relative mRNA levels were calculated using the *SCR1* reference gene. The data shows the mean ± SE (n = 5) of the fold change of *CLB1* mRNA relative to the mRNA level in wild-type. *P < 0.05, **P < 0.01 as determined by Tukey's test. (E, F) The *GFP* mRNA levels in wild-type, the *puf5Δ* mutant, the *ixr1Δ* mutant, and the *puf5Δ ixr1Δ* double mutant. The strains harboring the YCplac33-*CLB1* promoter-*GFP-ADH1* 3′ UTR plasmid (E) or YCplac33-*CLB2* promoter-*GFP-ADH1* 3′ UTR plasmid (F) were cultured in an SC-Ura medium at 28˚C until the exponential phase. The *GFP* mRNA levels were quantified by qRT-PCR analysis, and the relative mRNA levels were calculated using the *SCR1* reference gene. The data shows the mean ± SE (n = 3) of the fold change of *GFP* mRNA relative to the mRNA level in wild-type. *P < 0.05, **P < 0.01 as determined by Tukey's test.

*puf5Δ ixr1Δ* double mutant, the *puf5Δ* mutation decreased the endogenous *CLB1* and *GFP* mRNA levels even in the absence of Ixr1 (Fig 6D and 6E). These results imply that Puf5 regulates *CLB1* expression in both Ixr1-dependent and -independent manners.

Since the above data suggested that Ixr1 negatively regulates *CLB1* transcription, we questioned whether this regulation results from the direct binding of Ixr1 to the *CLB1* promoter.

However, since the interaction between Ixr1 and *CLB1* promoter has not been reported in a previous genome-wide chromatin immunoprecipitation (ChIP) analysis [43], it is unlikely that Ixr1 acts as a repressor that directly binds to the *CLB1* promoter. It is known that *CLB1* expression is induced by the activated Mcm1-Fkh2-Ndd1 complex [18,19]. Fkh1, a paralog of Fkh2, also regulates the transcription of *CLB1* and *CLB2* in the G2/M phase partially complementarily with Fkh2. Therefore, to consider a possibility that Ixr1 functions dependently on Fkh1 and Fkh2, we examined a genetic interaction between *IXR1* and *FKH1/FKH2*. We constructed the diploid strain heterozygous for *IXR1*, *FKH1*, and *FKH2* alleles and performed tetrad analysis. The *fkh1Δ* and *fkh2Δ* single mutants grew as well as wild-type (S4 Fig *fkh2Δ*, *fkh1Δ*, WT). In contrast, the *fkh1Δ fkh2Δ* double mutant showed a growth defect as previously reported [44]. This growth defect was not restored by *IXR1* deletion (S4 Fig *fkh1Δ fkh2Δ*, *ixr1Δfkh1Δ fkh2Δ*). These data implied the possibility that Fkh1 and Fkh2 may act downstream rather than parallel with Ixr1. Ixr1 may function via regulating expression of the transcriptional activators of *CLB1*. However, this result alone does not completely rule out the possibility that Fkh1 and Fkh2 regulate *CLB1* expression in parallel with Ixr1.

## Puf5 negatively regulates *IXR1* through the *IXR1* 3´ UTR

Ixr1 mediates the regulation of the *CLB1* expression by Puf5. As additional evidence, a previous genome-wide analysis reported that Puf5 binds to the 3´ UTR of *IXR1* mRNA [8]. Therefore, we investigated whether Puf5 regulates *IXR1* expression as a model shown in Fig 7A. The *IXR1* mRNA level in the *puf5Δ* mutant was 1.2-times as much as that in wild-type (Fig 7B). This difference was small, but reproducible. We also examined Ixr1 protein level. The level of Ixr1 protein tagged with HA in the *puf5Δ* mutant was 1.7-times as much as that in wild-type (Fig 7C and 7D). Since Puf5 has been reported to bind to the 3´ UTR of *IXR1* mRNA [8], we examined the effect of the *IXR1* 3´ UTR using the *IXR1*-HA-*IXR1* 3´ UTR and *IXR1*-HA-*ADH1* 3´ UTR constructs. The increase in Ixr1 protein level by *PUF5* deletion was significant in the construct harboring the *IXR1* 3´ UTR but not in the construct harboring the *ADH1* 3´ UTR (Fig 7C and 7D). These results implied that Puf5 negatively regulates *IXR1* expression via the *IXR1* 3´ UTR. In the *IXR1* 3´ UTR, we asked which region is responsible for the regulation by Puf5. Previous HITS-CLIP analysis has revealed that Puf5 binds to the element 5´-UGUAACAUUA in the 3´ UTR of *IXR1* mRNA [8]. We constructed YCplac33-*MCM2* promoter-*GFP*-*IXR1* 3´ UTR Δ UGUAACAUUA in which the Puf5-binding element is deleted (Fig 7E) and analyzed the effect of the deletion on GFP protein level. The deletion increased the GFP protein level 2.5-fold (Fig 7F and 7G), suggesting that Puf5-binding to the element negatively regulates expression of *IXR1* mRNA.

To further analyze the interaction between Puf5 and *IXR1* mRNA through the 3' UTR, we performed RNA immunoprecipitation (RIP). Puf5-FLAG protein was immunoprecipitated with anti-FLAG antibody, mRNAs were purified from the immunoprecipitants, and the enrichment of *IXR1* mRNA was analyzed (for details, see Materials and methods). Strains harboring untagged-Puf5 were used as a negative control. Western blot analysis was performed to confirm that Puf5-FLAG protein was contained in the supernatant and immunoprecipitants. While the bands of Leu1 protein recognized by the anti-FLAG M2 antibody were detected simultaneously, the bands of Puf5-FLAG protein were detected in the supernatant (Fig 8A). Although the bands of the Leu1 and Puf5-FLAG proteins was not separated in the immunoprecipitants, the signal intensity of the overlapping bands were clearly increased in the Puf5-FLAG strains compared to the untagged strain, indicating that the Puf5-FLAG protein was successfully immunoprecipitated (Fig 8E). Consistently, *SUN4* and *TOS1* mRNAs, which are reported to be strongly immunoprecipitated with Puf5 in the previous HITS-CLIP analysis [8],

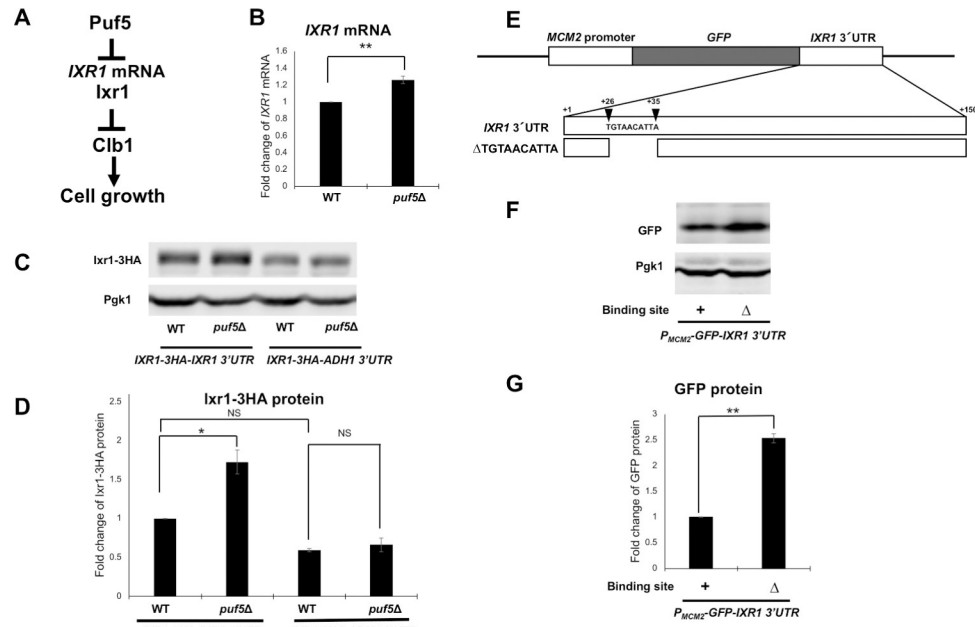

**Fig 7. Puf5 negatively regulates *IXR1* expression through the *IXR1* 3' UTR.** (A) A model showing how Puf5 functions in cell growth through the *CLB1* regulation, which is mediated by Ixr1. The hypothesis is presented that Puf5 negatively regulates the *IXR1* expression through the *IXR1* mRNA, and that Ixr1negatively regulates the *CLB1* transcription. (B) The *IXR1* mRNA levels in wild-type and the *puf5Δ* mutant. The cells were cultured in a YPD medium containing 10% sorbitol at 25°C until the exponential phase. The *IXR1* mRNA levels were quantified by qRT-PCR analysis, and the relative mRNA levels were calculated using the *SCR1* reference gene. The data shows the mean ± SE (n = 3) of the fold change of *IXR1* mRNA relative to the mRNA level in wild-type. *P < 0.05, **P < 0.01 as determined by Tukey's test. (C, D) The Ixr1 protein levels in wild-type and the *puf5Δ* mutant and the quantitative analysis data of Ixr1-HA protein level. The strains harboring the YCplac33-*IXR1*-HA-*IXR1* 3' UTR plasmid or the YCplac33-*IXR1*-HA-*ADH1* 3' UTR plasmid were cultured in an SC-Ura medium at 28°C. The extracts were immunoblotted with anti-HA antibody or anti-Pgk1 antibody. Ixr1-HA protein level was quantified and normalized with the Pgk1 protein level. The data shows the mean ± SE (n = 3) of the fold change of Ixr1-HA protein relative to the protein level in wild-type harboring the YCplac33-*IXR1*-HA-*IXR1* 3' UTR plasmid (D). *P < 0.05, **P < 0.01 as determined by Tukey's test. (E) Scheme of the YCplac33-*IXR1*-HA-*IXR1* 3' UTR plasmid. ΔTGTAACATTA harbors the deletion of the sequence encoding the Puf5-binding site of *IXR1* mRNA, 5'-TGTAACATTA. Described numbers correspond to the number of bases from the stop codon of *IXR1*. (F, G) The GFP protein levels and the quantitative analysis data of GFP protein level in wild-type strain compared by the presence of the Puf5-binding site deletion. The strains harboring the YCplac33-*MCM2* promoter-*GFP-IXR1* 3' UTR plasmid with or without the Puf5-binding site were cultured in an SC-Ura medium at 28°C. The extracts were immunoblotted with anti-GFP antibody or anti-Pgk1 antibody. GFP protein level was quantified and normalized with the Pgk1 protein level. The data shows the mean ± SE (n = 3) of the fold change of GFP protein relative to the protein level in the strains harboring the YCplac33-*MCM2* promoter-*GFP-IXR1* 3' UTR plasmid with the Puf5-binding site (G). *P < 0.05, **P < 0.01 as determined by Tukey's test.

were highly enriched in the Puf5-FLAG immunoprecipitants compared to the untagged immunoprecipitants (Fig 8B). Similarly, *IXR1* mRNA was significantly enriched in the Puf5-FLAG immunoprecipitants compared to the untagged immunoprecipitants (Fig 8B). We also investigated the contribution of Puf5-binding element to the enrichment of *IXR1* mRNA. The enrichment of *IXR1* mRNA was dependent on the Puf5-binding element (Fig 8B). In contrast, *SUN4* mRNA and *TOS1* mRNA enriched independently of the binding element in the *IXR1* 3' UTR (Fig 8B). These data confirmed that Puf5 specifically binds to the motif 5'-UGUAA-CAUUA in the *IXR1* 3' UTR.

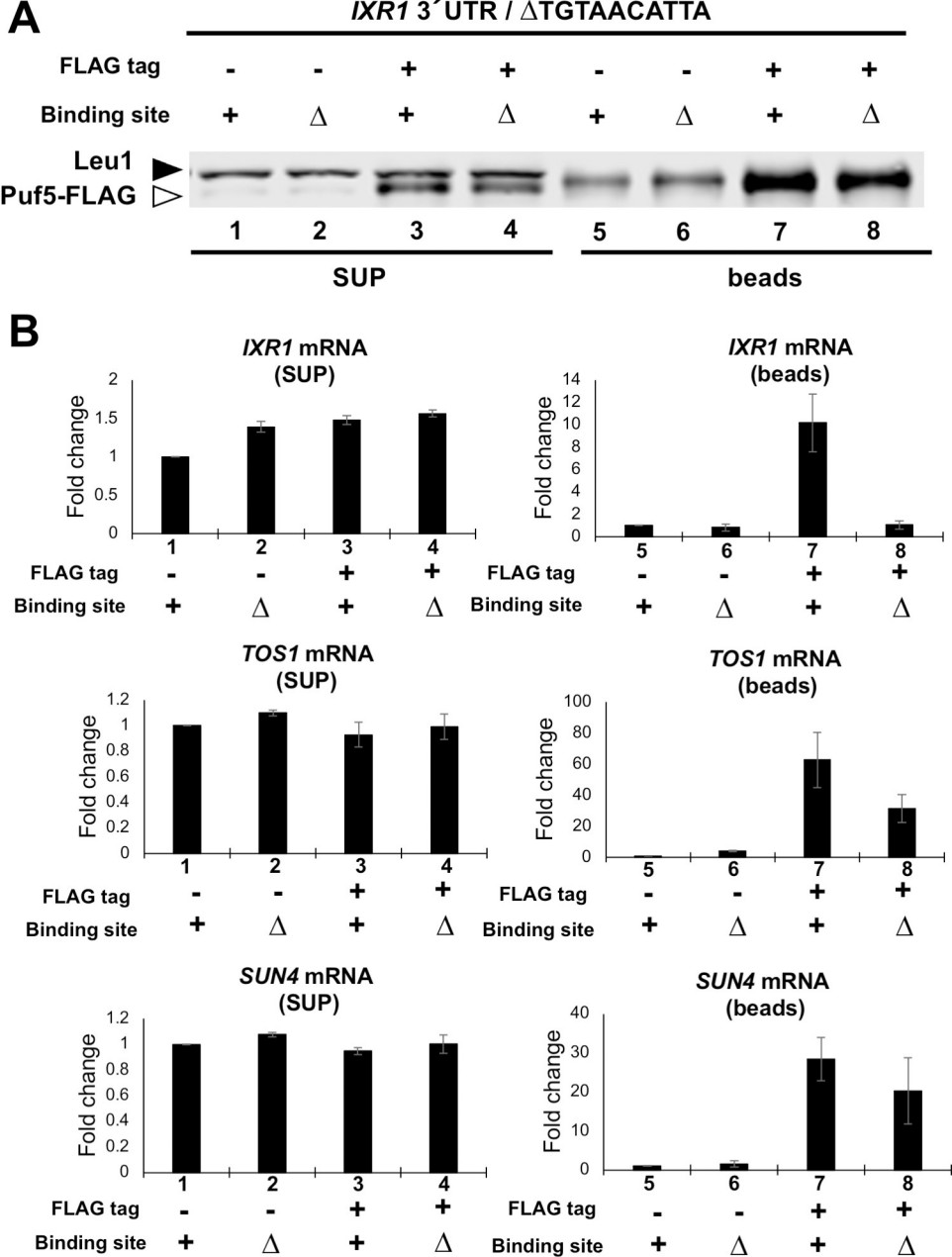

**Fig 8. Puf5 binds to 3´ UTR of *IXR1* mRNA.** (A,B) RIP analysis data clarifying the binding between Puf5 protein and *IXR1* mRNA. The extract was obtained from *ixr1Δ PUF5-FLAG-ADH1* 3´ UTR strain harboring the YEplac195-*IXR1* plasmid, and Puf5-FLAG protein was immunoprecipitated with anti-FLAG antibody. The *ixr1Δ* untagged-*PUF5* strain was used as a negative control. Sample 1–4 contain the supernatant, and sample 5–8 contain the immunoprecipitants. Sample 1 and 5, untagged strain with Puf5-binding element; sample 2 and 6, untagged strain without Puf5-binding element; sample 3 and 7, FLAG-tagged strain with Puf5-binding element; sample 4 and 8, FLAG-tagged strain without Puf5-binding element. (A) Puf5-FLAG protein level in supernatant and immunoprecipitants. The white arrowhead corresponds to Puf5-FLAG protein, and the black to Leu1 protein. (B) *IXR1* mRNA levels in the supernatant and the immunoprecipitants. *TOS1* mRNA levels and *SUN4* mRNA levels were presented as positive controls. The mRNA levels were quantified by qRT-PCR analysis, and the fold change was calculated relative to the mRNA level in the untagged strain with Puf5-binding element (sample 1 for the supernatant, and sample 5 for the immunoprecipitants).

## Puf5 and Ixr1 contribute to the cell-cycle-dependent expression of *CLB1*

The above data suggest that Puf5 regulates *CLB1* expression through regulating *IXR1* expression. However, when examining an asynchronous culture, the decrease of the *CLB1* expression in the *puf5Δ* mutant, approximately 50% decrease, was not prominent (Figs 2A, 6B and 6D). Regarding that the expression of *CLBs* dramatically transits depending on the cell cycle [16], the decrease of the *CLB1* expression in the *puf5Δ* mutant may be observed at a higher extent in a synchronous culture. Therefore, we synchronized the cell cycle by pheromone-induced G1 arrest and examined the effects of Puf5 and Ixr1 on the cell cycle-specific expression of *CLB1* mRNA. For this purpose, we utilized the *MAT***a** *bar1Δ* and *MAT***a** *bar1Δ puf5Δ* cells, in which α-factor protease Bar1 was absent [27]. The *MAT***a** *bar1Δ* and *MAT***a** *bar1Δ puf5Δ* cells were arrested in the G1 phase with α-factor and synchronized by releasing, and RNAs were extracted from the samples collected from 0 minutes (just before releasing) to 150 minutes (for details, see Materials and methods). We examined the expression of *RNR1* and *SIC1* in cell cycle-synchronized strains as an S phase and a late M phase marker, respectively. In the *bar1Δ* cell, the *RNR1* mRNA level peaked at 40 min first and at 90–100 min second (Fig 9A *bar1Δ*), and the *SIC1* mRNA level did at 80 min first and at 140 min second (Fig 9B *bar1Δ*). In the *bar1Δ puf5Δ* mutant, the *RNR1* mRNA level reached the summit at 50 min first and at 120 min second (Fig 9A *bar1Δ puf5Δ*), and the *SIC1* mRNA level did at 100-120 min (Fig 9B *bar1Δ puf5Δ*). These data show that the cell cycle was completed within the time tested in both strains, although the cell cycle was delayed for 10–40 min in the *bar1Δ puf5Δ* mutant. The expression of *CLB1* mRNA was induced from 40 min, peaked at 60 min, and decreased immediately after the peak in the *bar1Δ* cell (Fig 9C *bar1Δ*). As expected, the *CLB1* mRNA level was dramatically decreased in the *bar1Δ puf5Δ* mutant: the expression of *CLB1* was slightly induced at 80-90 min, but the extent of the induction was much smaller in the *bar1Δ puf5Δ* mutant than that in the *bar1Δ* cell (Fig 9C *bar1Δ*, *bar1Δ puf5Δ*). These results suggest that Puf5 ensures the utmost expression of *CLB1* during the cell cycle through positive regulation of *CLB1* expression.

Regarding that Ixr1 mediates the regulation of *CLB1* expression by Puf5, we examined whether the *ixr1Δ* mutation restored the decreased expression of *CLB1* in the synchronous *puf5Δ* mutant. The *CLB1* mRNA levels in the synchronous *bar1Δ* cell and *bar1Δ puf5Δ ixr1Δ* mutant was compared to that in *bar1Δ puf5Δ* mutant shown in Fig 9C. The expression of *RNR1* mRNA, an S-phase marker, showed similar patterns in the three strains and peaked at 40–50 min (Fig 10A *bar1Δ*, *bar1Δ puf5Δ*, *bar1Δ puf5Δ ixr1Δ*). This result suggests that the cell cycle progresses normally in all three strains. As shown in Fig 9C, the cell cycle-specific expression of *CLB1* mRNA was remarkably diminished in the *bar1Δ puf5Δ* mutant compared to that in the *bar1Δ* cell (Fig 10B *bar1Δ*, *bar1Δ puf5Δ*), and the diminished expression was restored in the *bar1Δ puf5Δ ixr1Δ* mutant (Fig 10B *bar1Δ puf5Δ ixr1Δ*). These results indicate that Ixr1 functions downstream of Puf5 and mediates the positive control of the cell cycle-specific *CLB1* expression. To confirm the involvement of Ixr1 in cell cycle progression, we also examined the *CLB1* mRNA level in the cell cycle-synchronized *bar1Δ ixr1Δ* mutant. The peak of *RNR1* mRNA level was observed at 40 min in the *bar1Δ* cell and 50 min in the *bar1Δ puf5Δ* mutant as shown in Fig 9A and 10A (Fig 11A *bar1Δ*, *bar1Δ puf5Δ*). In the *bar1Δ ixr1Δ* mutant, the *RNR1* mRNA level went highest at 20 min first, approximately 20 minutes earlier than in the *bar1Δ* cell, and at 80 min second (Fig 11A *bar1Δ ixr1Δ*). The extent of the peak expression was similar to that in the *bar1Δ* cell (Fig 11A *bar1Δ*, *bar1Δ ixr1Δ*). Although the expression of *RNR1* was induced earlier than in *bar1Δ* cell, these data suggest that the cell cycle progresses regularly in the *ixr1Δ* background. As shown in Fig 10B, the *CLB1* mRNA level was highest around 60–80 min in the *bar1Δ* cell, and the peak was remarkably lessened in the *bar1Δ puf5Δ*

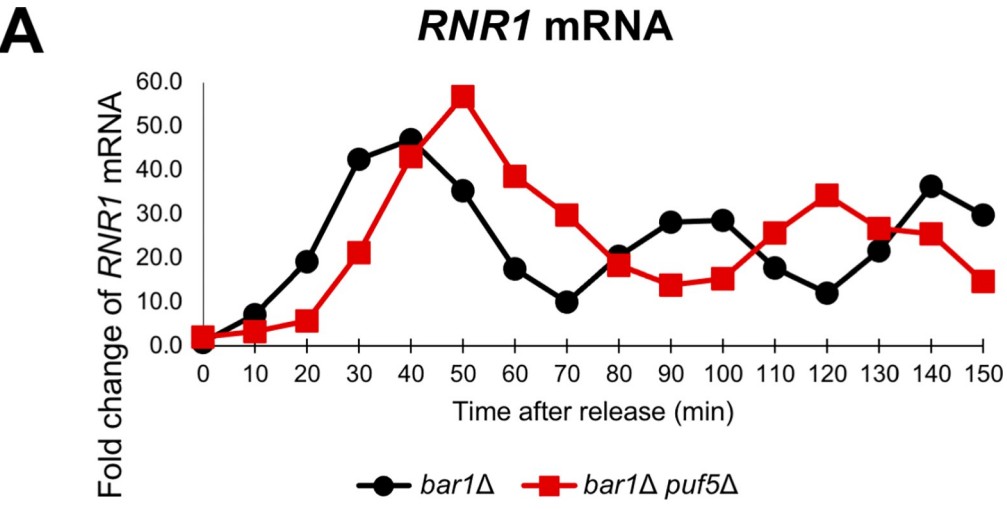

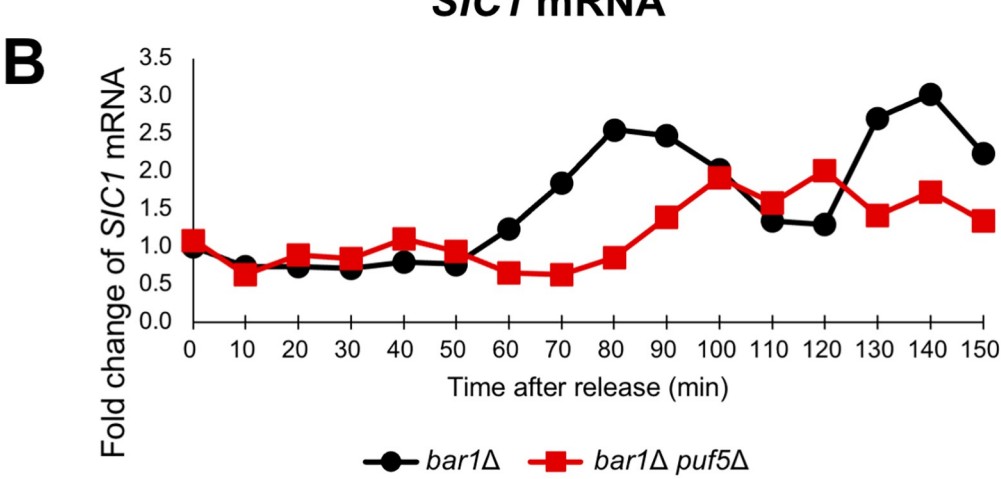

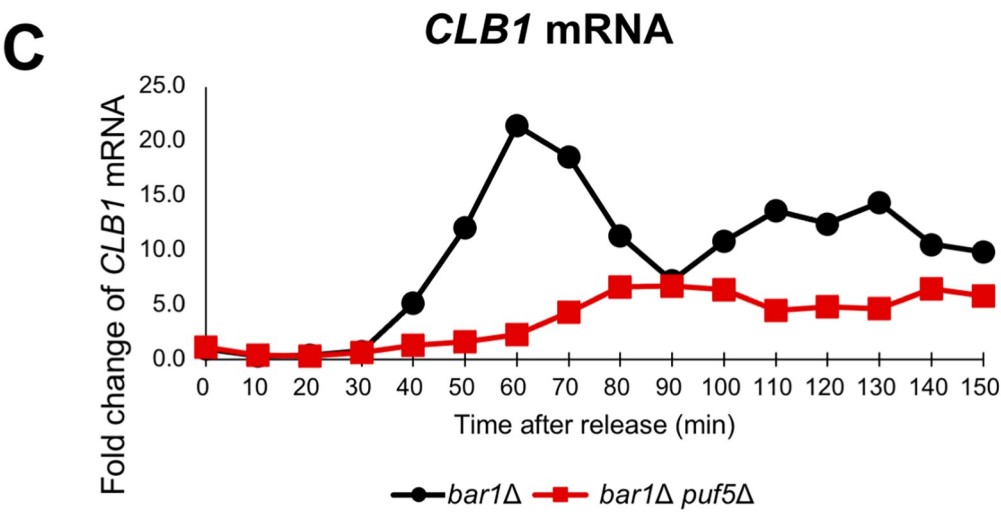

**Fig 9. The cell cycle-regulated expression of *CLB1* was diminished in the *puf5*Δ mutant.** (A-C) The cell cycle-dependent mRNA levels of *CLB1* in the synchronized *bar1*Δ cell (black circle) and *bar1*Δ *puf5*Δ mutant (red square). The cell cycle was arrested in the G1 phase by α-factor, and, after release, cells were collected from 0 min (just before releasing) to 150 min. The levels of *RNR1* mRNA (A), an S phase marker, *SIC1* mRNA (B), a late M phase marker, and *CLB1* mRNA (C) were quantified by qRT-PCR analysis, and the relative mRNA levels were calculated using the *SCR1* reference gene. The vertical axis shows the fold change of mRNA relative to the mRNA level in the *bar1*Δ 0 min sample, and the horizontal axis shows the time after release.

mutant (Fig 11B *bar1*Δ, *bar1*Δ *puf5*Δ). Comparing the *bar1*Δ cell and the *bar1*Δ *ixr1*Δ mutant, the expression of *CLB1* was induced at an earlier stage, and the extent of the peak expression was significantly higher in the *bar1*Δ *ixr1*Δ mutant than in the *bar1*Δ cell (Fig 11B *bar1*Δ, *bar1*Δ *ixr1*Δ). These data suggest that Ixr1 negatively regulates cell cycle-depending induction of *CLB1* expression. Puf5, a negative regulator of *IXR1* expression, seems to control this Ixr1-mediated repression and indirectly control the maximum level of the cell cycle-dependent *CLB1* expression, which probably leads to the control of the proper cell cycle progression.

## Puf5 negatively regulates *IXR1* expression throughout the cell cycle

Regarding that Puf5 and Ixr1 seem to regulate the cell cycle progression, we questioned whether their expression rhythmically transits during the cell cycle. We first examined *PUF5* expression in the cell cycle. *PUF5* mRNA levels in synchronized *bar1*Δ cells did not change significantly throughout the cell cycle (Fig 12A). Consistent with the mRNA level, Puf5-Myc protein levels did not change significantly throughout the cell cycle examined using the *bar1*Δ *PUF5*-Myc strain (Fig 12B). Next, we analyzed the transition of *IXR1* expression in a synchronous culture. *IXR1* mRNA levels in both the *bar1*Δ cell and the *bar1*Δ *puf5*Δ mutant did not change significantly throughout the cell cycle (Fig 13A *bar1*Δ, *bar1*Δ *puf5*Δ). Even though the periodic expression of *IXR1* mRNA was not observed, the *IXR1* mRNA level was higher in the *bar1*Δ *puf5*Δ mutant than that in the *bar1*Δ cell over the cell cycle (Fig 13A *bar1*Δ, *bar1*Δ *puf5*Δ). The Ixr1 protein level was quantified by exploiting the YCplac33-*IXR1*-HA-*IXR1* 3′ UTR plasmid in the *bar1*Δ cell and the *bar1*Δ *puf5*Δ mutant. Since the plasmids were used, we first examined the *IXR1*-HA mRNA level; the *IXR1*-HA mRNA level in the *bar1*Δ cell did not change significantly during the cell cycle, consistently with the endogenous *IXR1* mRNA level shown in Fig 13A (Fig 13B *bar1*Δ). In the *bar1*Δ *puf5*Δ mutant, the *IXR1*-HA mRNA level was increased and went higher following the cell cycle progression than that in the *bar1*Δ cell (Fig 13B *bar1*Δ *puf5*Δ, *bar1*Δ). Similar to the *IXR1*-HA mRNA level, the Ixr1 protein level in the *bar1*Δ cell did not change significantly during the cell cycle (Fig 13C *bar1*Δ). In the *bar1*Δ *puf5*Δ mutant, the Ixr1 protein level was increased following the cell cycle progression than that in the *bar1*Δ cell (Fig 13C *bar1*Δ *puf5*Δ, *bar1*Δ). These results suggest that expression of Puf5 and Ixr1 does not change significantly during the cell cycle, but Puf5 negatively regulates *IXR1* expression throughout the cell cycle.

## Discussion

### Puf5 contributes to cell growth through the positive regulation of the *CLB1* expression, which is mediated by Ixr1

Puf5 has been reported to play multiple roles including mating-type switching, maintaining CWI, and extending life span [3,5,8–15]. Here, we identified *CLB2* as a multicopy suppressor of the temperature-sensitive phenotype of the *puf5*Δ mutant. To reveal the machinery for the suppression by *CLB2*, we further analyzed the Puf5 function in cell growth using the genetic approaches. We found that Puf5 positively regulates the *CLB1* transcription, and that this regulation has a phenotypic effect in the absence of Clb2. For the mechanism of how Puf5 regulates

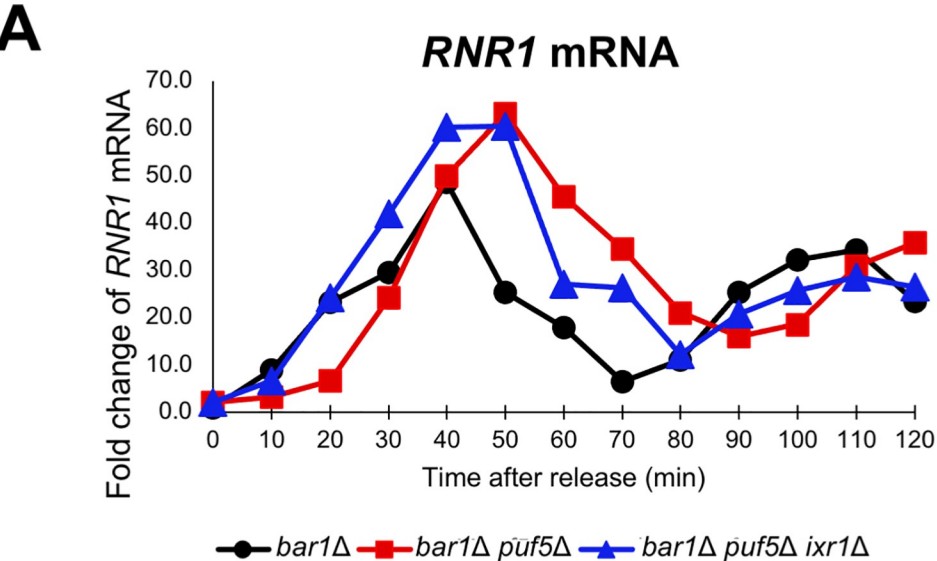

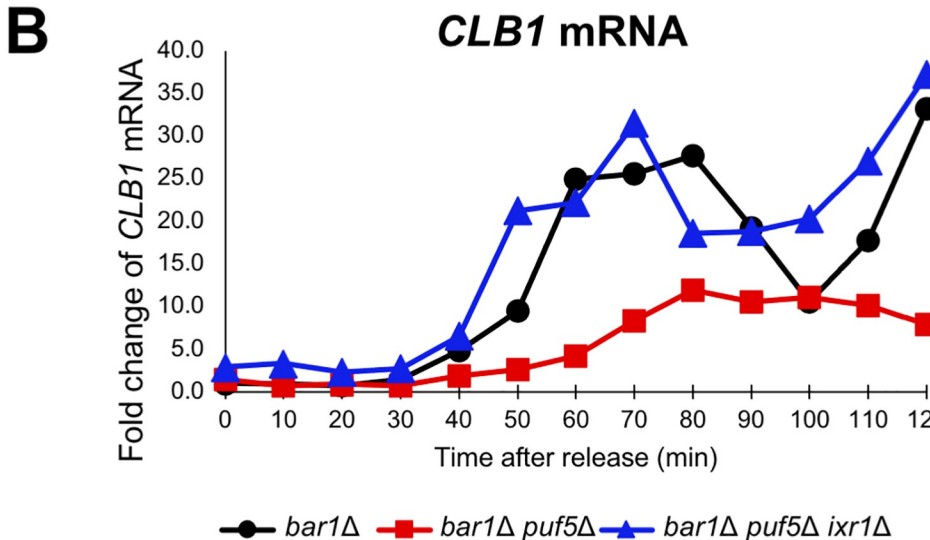

**Fig 10. *IXR1* deletion restored the decreased expression of *CLB1* caused by *PUF5* deletion.** (A, B) The cell cycle-dependent mRNA levels of *CLB1* (B) in the cell cycle synchronized *bar1Δ* cell (black circle), *bar1Δ puf5Δ* mutant (red square), and *bar1Δ puf5Δ ixr1Δ* mutant (blue triangle). The cell cycle was arrested in the G1 phase by α-factor, and, after release, cells were collected from 0 min (just before releasing) to 120 min. The levels of *RNR1* mRNA (A), an S phase marker, and *CLB1* mRNA (B) were quantified by qRT-PCR analysis, and the relative mRNA levels were calculated using the *SCR1* reference gene. The vertical axis shows the fold change of mRNA relative to the mRNA level in the *bar1Δ* 0 min sample, and the horizontal axis shows the time after release.

the *CLB1* transcription, we hypothesized that Puf5 negatively regulates the expression of a factor that negatively regulates the transcription of *CLB1*, since Puf5 has been reported to negatively regulate the expression of target mRNAs via their 3′ UTRs [6,7,10–13]. We identified such a factor Ixr1. Preceding genome-wide RIP analysis data have revealed the interaction of Puf5 with cyclin mRNAs, *CLB1*, *CLB2*, *CLB3*, and *CLN1* mRNA [8,9]. Among them, only *CLB1* expression was affected by *puf5Δ* mutation (Table 1). Regardless of the protein-RNA

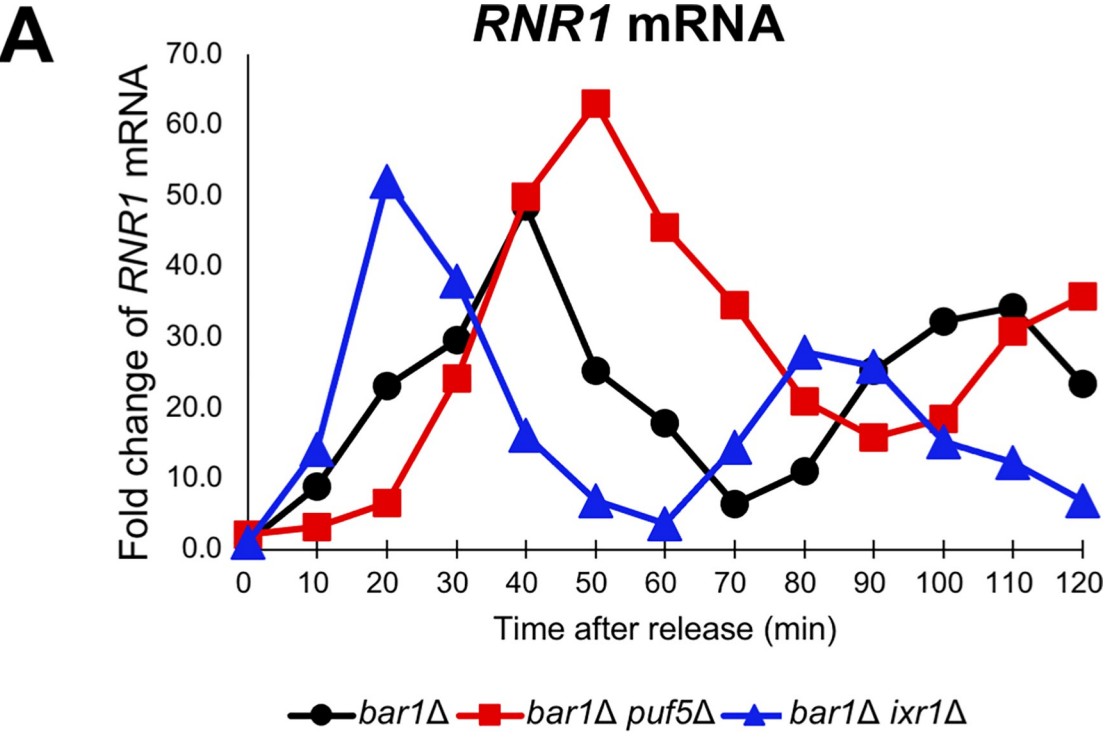

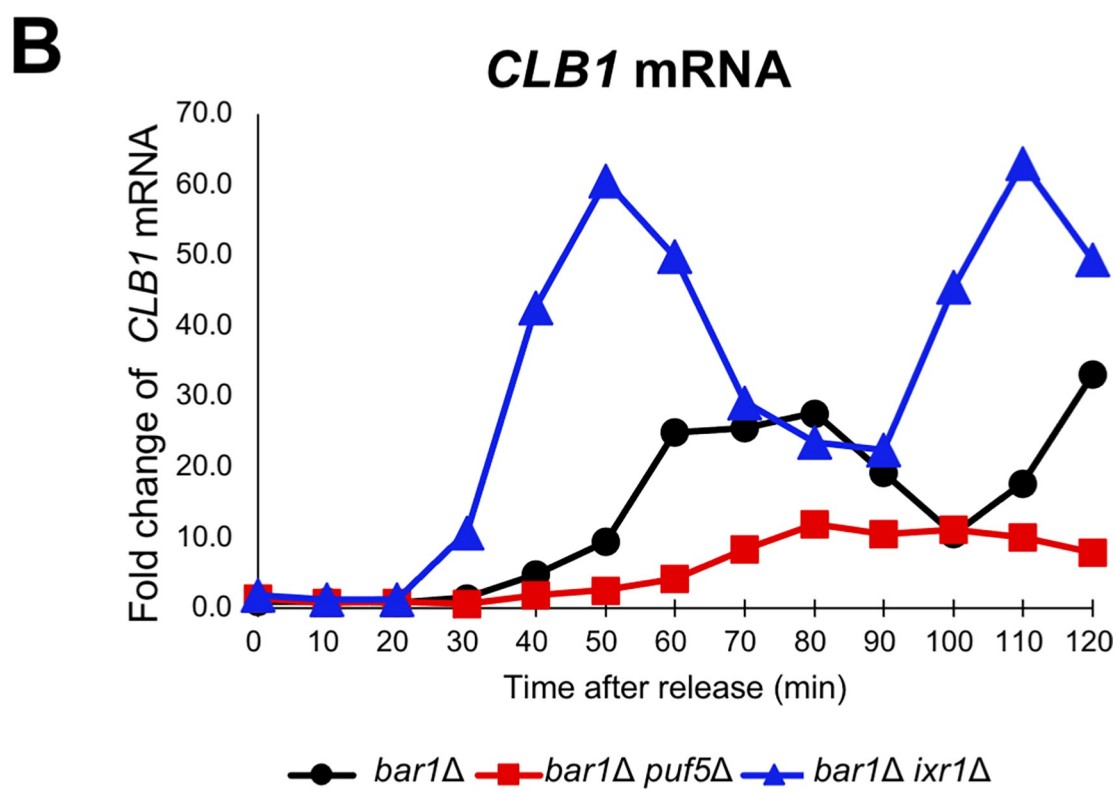

**Fig 11. Ixr1 negatively regulates the cell cycle-specific expression of *CLB1*.** (A, B) The cell cycle-dependent mRNA levels of *CLB1* (B) in the synchronized *bar1Δ* cell (black circle), *bar1Δ puf5Δ* mutant (red square), and *bar1Δ ixr1Δ* mutant (blue triangle). The cell cycle was arrested in the G1 phase by α-factor, and, after release, cells were collected from 0 min (just before releasing) to 120 min. The levels of *RNR1* mRNA (A), an S phase marker, and *CLB1* mRNA (B) were quantified by qRT-PCR analysis, and the relative mRNA levels were calculated using the *SCR1* reference gene. The vertical axis shows the fold change of mRNA relative to the mRNA level in the *bar1Δ* 0 min sample, and the horizontal axis shows the time after release.

interaction, Puf5 indirectly regulates *CLB1* expression: Puf5 negatively regulates expression of *IXR1* encoding a transcriptional repressor with the HMG domain [34,35]. Previous microarray analysis has reported that *CLB1* expression is increased in the *ixr1Δ* mutant [42], but we first

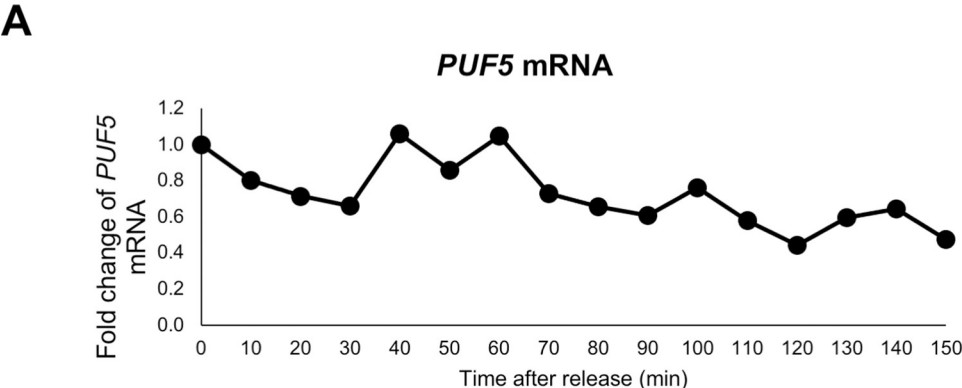

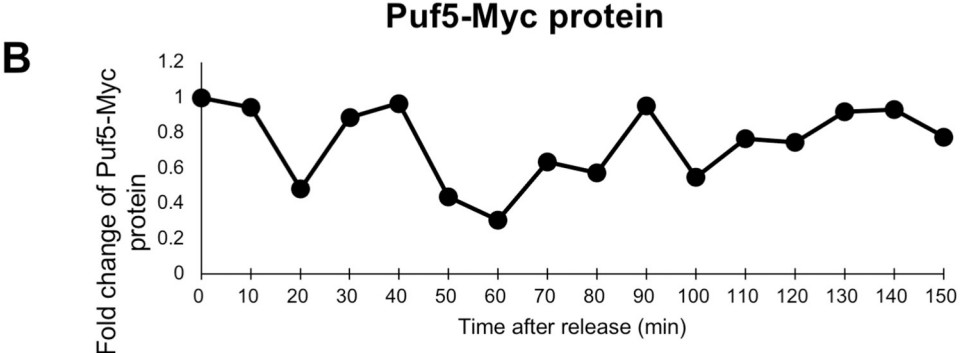

**Fig 12. *PUF5* expression is invariable during the cell cycle.** (A) The cell cycle-dependent mRNA levels of *PUF5* in the synchronized *bar1Δ* cell, the same sample used in Fig 9. The cell cycle was arrested in the G1 phase by α-factor, and, after release, cells were collected from 0 min (just before releasing) to 150 min. The levels of *PUF5* mRNA were quantified by qRT-PCR analysis, and the relative mRNA levels were calculated using the *SCR1* reference gene. The vertical axis shows the fold change of mRNA relative to the mRNA level in the *bar1Δ* 0 min sample, and the horizontal axis shows the time after release. (B) The cell cycle-dependent Puf5 protein level in the synchronized *bar1Δ* cell integrated *PUF5-13Myc-ADH1* 3′ UTR gene. The cell cycle was arrested in the G1 phase by α-factor, and, after release, cells were collected from 0 min (just before releasing) to 150 min. The extracts were immunoblotted with anti-Myc antibody or anti-Pgk1 antibody. The blot image is presented in S5A Fig. Puf5-Myc protein level was quantified and normalized with the Pgk1 protein level. The vertical axis shows the fold change of protein relative to the protein level in the *bar1Δ* 0 min sample, and the horizontal axis shows the time after release.

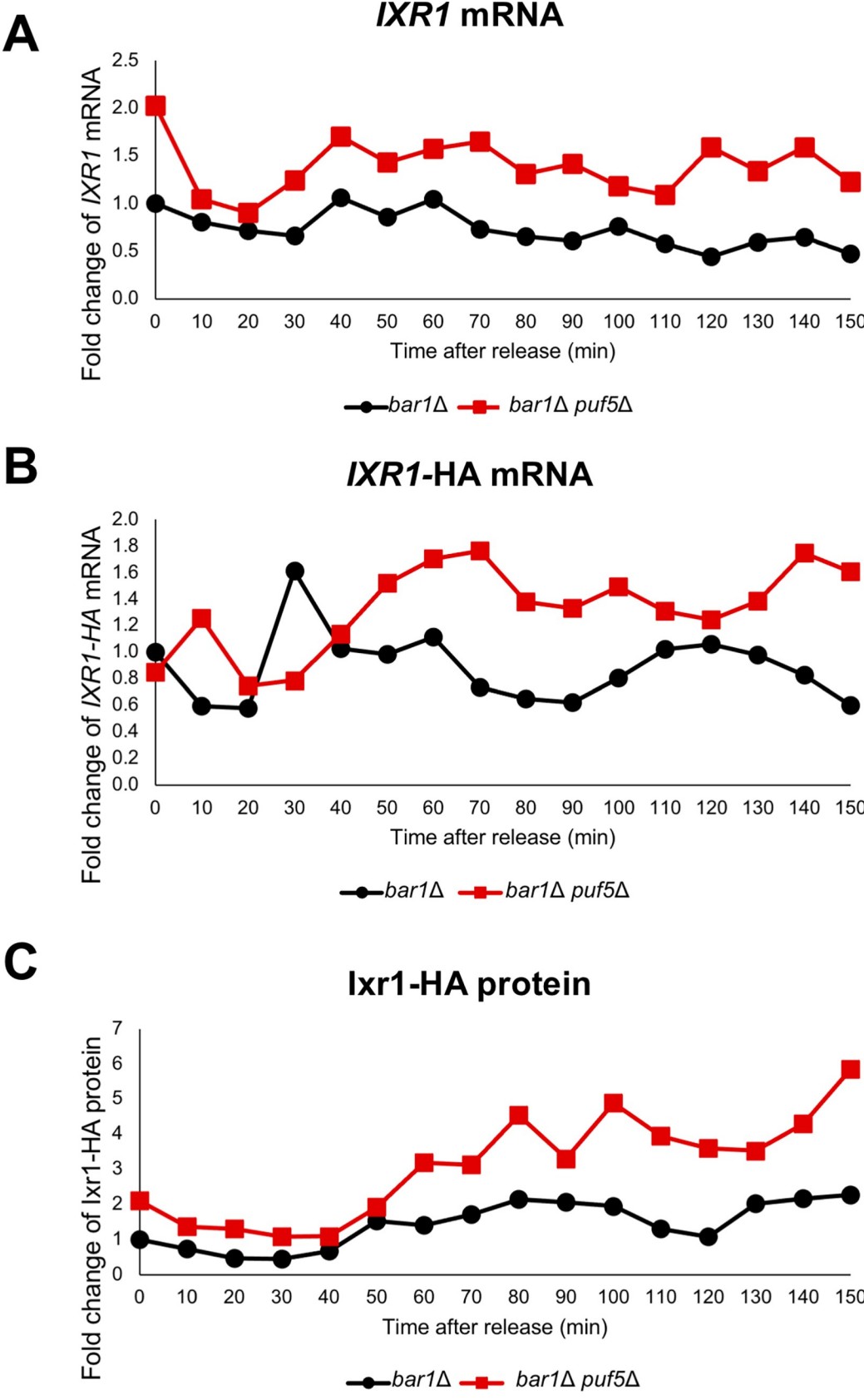

**Fig 13. *IXR1* expression is invariable during the cell cycle.** (A) The *IXR1* mRNA levels in the synchronized *bar1Δ* cell (black circle) and *bar1Δ puf5Δ* mutant (red square), the same sample used in Fig 9. The cell cycle was arrested in the G1 phase by α-factor, and, after release, cells were collected from 0 min (just before releasing) to 150 min. The *IXR1* mRNA level was quantified by qRT-PCR analysis, and the relative mRNA levels were calculated using the *SCR1* reference gene. The mRNA level in the *bar1Δ* 0 min sample was set as a reference. The vertical axis shows the fold change of *IXR1* mRNA relative to the mRNA level in the *bar1Δ* 0 min sample, and the horizontal axis shows the time after release. (B) The *IXR1*-HA mRNA levels in the synchronized *bar1Δ* cell (black circle) and *bar1Δ puf5Δ* mutant (red square) harboring the YCplac33-*IXR1*-HA-*IXR1* 3′ UTR plasmid. The cell cycle was arrested in the G1 phase by α-factor, and, after release, cells were collected from 0 min (just before releasing) to 150 min. The *IXR1*-HA mRNA level was quantified by qRT-PCR analysis, and the relative mRNA levels were calculated using the *SCR1* reference gene. The mRNA level in the *bar1Δ* 0 min sample was set as a reference. The vertical axis shows the fold change of *IXR1*-HA mRNA relative to the mRNA level in the *bar1Δ* 0 min sample, and the horizontal axis shows the time after release. (C) The Ixr1-HA protein level in the synchronized *bar1Δ* cell (black circle) and *bar1Δ puf5Δ* mutant (red square) harboring the YCplac33-*IXR1*-HA-*IXR1* 3′ UTR plasmid. The cell cycle was arrested in the G1 phase by α-factor, and, after release, cells were collected from 0 min (just before releasing) to 150 min. The extracts were immunoblotted with anti-HA antibody or anti-Pgk1 antibody. The blot image is presented in S5B–S5D Fig. Ixr1-HA protein level was quantified and normalized with the Pgk1 protein level. The Ixr1-HA protein level in the *bar1Δ* 0 min sample was set as a reference. The vertical axis shows the fold change of protein relative to the protein level in the *bar1Δ* 0 min sample, and the horizontal axis shows the time after release.

clarified that Ixr1 negatively regulates *CLB1* transcription. Together with our data that the *ixr1Δ* mutation recovered the growth defect of *puf5Δ clb2Δ* double mutant, Ixr1 negatively regulates the *CLB1* transcription with physiological importance. However, it remains unclear whether Ixr1 represses the *CLB1* transcription directly or indirectly, by binding to the *CLB1* promoter or by controlling other regulatory factors of *CLB1*. Previous genome-wide ChIP analysis data did not report the binding of Ixr1 to the *CLB1* promoter [43]. Ixr1 has been reported to contribute to the transcriptional regulation of ribosomal genes [43]. Moreover, the association of Ixr1 with Ssn8 and Tdh3, general transcriptional regulators, was also reported [45]. Taken together, Ixr1 may function as a general repressor. For another possibility, Ixr1 may regulate the expression of the transcriptional activators of *CLB1*. The *CLB1* expression is induced by the activated Mcm1-Fkh2-Ndd1 complex [18,19,36,46]. Our result suggests that Fkh1/Fkh2 may function downstream of Ixr1 (S4 Fig), but this result does not completely rule out the possibility that Fkh1/Fkh2 functions parallel to Ixr1. Further analysis is needed to reveal the detailed machinery of the Ixr1 function.

### *IXR1* mRNA is a physiologically important target of Puf5

Puf5 negatively regulates *IXR1* expression, which leads to positive regulation of *CLB1* expression. This machinery of regulating *IXR1* expression depends on the Puf5-binding element in the *IXR1* 3′ UTR (Fig 7F and 7G). Furthermore, the interaction between Puf5 protein and *IXR1* mRNA through the *IXR1* 3′ UTR was confirmed (Fig 8). Taking these data into account, *IXR1* mRNA is a physiologically important target of Puf5. Considering the mechanism of how Puf5 regulates *IXR1* expression, the increase of *IXR1* expression in the *puf5Δ* mutant was observed more significantly in the protein level (approximately 1.7-fold) than in the mRNA level (1.2-fold) (Fig 7). Therefore, the negative regulation of *IXR1* mRNA by Puf5 may be caused by the translational control rather than by the mRNA stability control. Previously, it has been reported that Puf5 functions in the translational repression through binding to 3′ UTR of target mRNAs in vitro [47]. Regarding that, Puf5 may repress the translation of *IXR1* mRNA depending on the *IXR1* 3′ UTR.

### Puf5 regulates the expression of physiological targets, *IXR1* and *LRG1*, independently

The first observation that *CLB2* is a multicopy suppressor of the *puf5Δ* mutant led us to the finding of the Puf5 function in the *CLB1* regulation. However, since multicopy *CLB2*

suppressed the phenotype at 35˚C but not enough at 37˚C (Fig 1A), *CLB2* seems to weakly suppress the phenotype. It appears that other factors also contribute to the defect of the *puf5Δ* mutant.

We and other groups have previously reported that Puf5 negatively regulates the expression of *LRG1* encoding a Rho1 GAP involved in maintaining CWI [11–13,15]. This negative regulation involves *CCR4* and *POP2*, and the *puf5Δ* mutant, the *ccr4Δ* mutant, and the *pop2Δ* mutant show the cell lysis phenotype at the elevated temperature [11–13,15]. The *lrg1Δ* mutation partially suppresses the cell lysis phenotype, while *LRG1* overexpression exacerbates the cell lysis phenotype. In the regulation of the *CLB1* expression, Puf5 seems to function independently of controlling CWI through the *LRG1* regulation: the growth defect of the *puf5Δ clb2Δ* double mutant was not recovered by the *lrg1Δ* mutation (Fig 3E). In addition, *IXR1* deletion restored the growth of the *puf5Δ clb2Δ* double mutant even in the *lrg1Δ* background (Fig 5B). While *IXR1* deletion suppressed the temperature-sensitive phenotype of the *puf5Δ* mutant (Fig 5F), *IXR1* deletion did not retrieve the temperature sensitiveness of the *puf5Δ clb2Δ* double mutant (S3 Fig). The *lrg1Δ* mutation recovered the growth defect of the *puf5Δ clb2Δ ixr1Δ* triple mutant at the high temperature (S3 Fig). This maybe because the two physiologically important targets, *IXR1* and *LRG1*, independently act on cell growth. The Ixr1-mediated decrease of *CLB1* expression in the *puf5Δ* mutant partially causes the temperature-sensitive phenotype, and this phenotype is severely accelerated when the *clb2Δ* mutation is combined. Since the *lrg1Δ* mutation that partially suppresses the cell lysis phenotype affects cell growth independently of the Ixr1-mdiated *CLB1* regulation, the mutation is able to recover the growth defect of the *puf5Δ clb2Δ* double mutant at the elevated temperature.

## The cell cycle-specific expression of *CLB1* is regulated by not only positive regulators but also negative regulators

Expression of cyclin genes dramatically transits depending on the cell cycle [16]. Here we found that Puf5 and its physiological target Ixr1 are involved in controlling the cell cycle-specific expression of *CLB1*, which leads to the control of the proper cell cycle progression. When examined in a synchronous culture, the cell cycle was delayed for approximately 10–20 min in the *puf5Δ* background (Fig 9) and started to progress at an earlier stage in the *ixr1Δ* background (Fig 11). We think of two hypotheses for the observation. (a) Puf5 and Ixr1 are involved in pheromone recovery, since we synchronized the cell cycle using pheromone-induced cell cycle arrest. (b) Puf5 and Ixr1 generally regulate the cell cycle progression. Further analyses are needed for the detailed mechanism. Previous study has revealed that Ixr1 positively regulates the expression of *RNR1* [48]. However, from our data, the *RNR1* expression was induced earlier in the *ixr1Δ* mutant, and the *ixr1Δ* mutation did not affect the extent of the peak expression. These data are not consistent with the previous report [48]. The difference of the yeast strains may result in this inconsistency.

Puf5 and its target Ixr1 regulate the *CLB1* expression specifically in G2/M phase. Therefore, there should be the regulatory machinery of the cell cycle-depending activity of Puf5 and Ixr1. However, the Puf5 and Ixr1 mRNA levels were invariable through the cell cycle (Figs 12 and 13). Similarly, our western blot data examining Puf5 or Ixr1 protein levels during the cell cycle showed plural bands of each protein (S5A–S5D Fig). Therefore, cell cycle-regulated modification, such as phosphorylation, of Puf5 and Ixr1 protein may determine their phase-specific activity. Further analyses are required to explore the mechanism of how the activities of Puf5 and Ixr1 are rhythmically regulated and how the rhythmicity influences the *CLB1* expression.

The cell cycle-dependent transcription of *CLB1*/*CLB2* has been reported to be regulated by the periodic activity of the Mcm1-Fkh2-Ndd1 activator complex [18,19,36,46]. Even though

the importance of the transcriptional activators has been reported, little has been reported about the contribution of a transcriptional repressor to the regulation of the G2/M phase transcription. Our data first revealed that Ixr1 transcriptionally represses the cell cycle-specific expression of *CLB1*, although the detailed mechanism of the repression is unclear. Here we identified Puf5 and Ixr1 as cell cycle regulators. Ixr1 represses the G2/M phase-specific expression of *CLB1*, and Puf5 positively regulates the expression through the negative regulation of the *IXR1* expression. This machinery ensures the appropriate expression of *CLB1* and sequentially contributes to the proper cell cycle progression. Regarding how this regulation administers to the cell cycle regulation, we consider that this regulation is involved in the G2/M phase checkpoint. In *S. cerevisiae*, DNA damage and DNA replication failure provoke the G2/M phase checkpoint [49]. *IXR1* gene has been reported to interact with *RAD9* gene encoding an adaptor protein involved in DNA damage checkpoint [50]. Moreover, the *ixr1Δ* mutation improves cell growth when replication stress is induced [51]. Ixr1 indirectly positively regulates the dNTP pool [48], and the decreased concentration of dNTP accommodates cells to the replication stressed condition [51]. Taking together, Ixr1 may be a downstream factor of the DNA damage or DNA replication checkpoint. If this hypothesis is true, the activity of Ixr1 controlled by the checkpoint alters the amount of *CLB1* transcripts, leading to the appropriate cell division. In humans, HMGB proteins also play a role in DNA repair, and increased expression of HMGB1, one type of HMGB protein, is associated with tumor progression or metastasis [52], suggesting that HMGB proteins are closely related to maintaining ordered cell proliferation.

### Insights into RNA regulatory network

In the regulation of cell cycle progression, the importance of the post-transcriptional regulation is less investigated. Previously, RNA-binding protein Whi3 has been reported to interact with *CLN3* mRNA [53,54], and, more recently, we have shown that Ccr4, a catalytic subunit of Ccr4-Not complex, has been found to destabilize *CLB6* mRNA [55]. However, it has been unclear whether RNA-binding proteins contribute to the regulation of the expression of B-type cyclins. In this study, we found that Puf5-binding to the 3′ UTR of *IXR1* mRNA leads to the negative regulation of *IXR1* expression, which results in the positive regulation of *CLB1* expression and control of the cell cycle progression. These findings indicate the significance of the RNA regulation upon cell cycle control. In other species, such as mouse, *Xenopus*, and zebrafish, PUF family protein Pum1 directly binds to cyclin B1 mRNA and negatively regulates the translation of cyclinB1 in oocytes [56–59]. Contrarily, in *S. cerevisiae*, Puf5 indirectly regulates the expression of *CLB1*, a cyclin B gene, via the regulation of the expression of *IXR1*. This difference shows a variety of the function of PUF family proteins acquired during evolution. Puf5 functions in tolerating various stress, such as replication stress, weakened cell wall, and high temperature, and positively regulates life span [11–15]. Such pleiotropically functioning Puf5 controls the appropriate *CLB1* expression level at the G2/M phase or upon DNA damage, suggesting that Puf5 controls the cell cycle progression in a proper state for the circumstances. This regulation may enhance the fitness of cells and prolong the life span.

### Supporting information

**S1 Table. Strains used in this study.**
(DOCX)

**S2 Table. Plasmids used in this study.**
(DOCX)

**S3 Table. Primers used for the gene deletion.**
(DOCX)

**S4 Table. Primers used for the qRT-PCR.**
(DOCX)

**S5 Table. Fold change of the mRNA level of regulators of *CLB1*.**
(DOCX)

**S1 Data. Numerical data of the graphs.**
(XLSX)

**S1 Fig. *ADH1* is used as a control gene whose expression is not affected by *PUF5* or *IXR1* deletion.** (A) The mRNA levels of *ADH1* in wild-type, the *puf5*Δ mutant, the *ixr1*Δ mutant, and the *puf5*Δ *ixr1*Δ double mutant. The cells were cultured in a YPD medium at 28°C until the exponential phase. The *ADH1* mRNA levels were quantified by qRT-PCR analysis, and the relative mRNA levels were calculated using the *ACT1* reference gene. The data shows the mean ± SE (n = 3) of the fold change of *ADH1* mRNA relative to the mRNA level in wild-type. $^*P < 0.05$, $^{**}P < 0.01$ as determined by Tukey's test. (B) The mRNA levels of *CLB2* in wild-type, the *puf5*Δ mutant, the *ixr1*Δ mutant, and the *puf5*Δ *ixr1*Δ double mutant. The cells were cultured in a YPD medium at 28°C until the exponential phase. The *CLB2* mRNA levels were quantified by qRT-PCR analysis, and the relative mRNA levels were calculated using the *SCR1* reference gene. The data shows the mean ± SE (n = 3) of the fold change of *CLB2* mRNA relative to the mRNA level in wild-type. $^*P < 0.05$, $^{**}P < 0.01$ as determined by Tukey's test. (TIFF)

**S2 Fig. Screening of downstream factor of Puf5.** (A) The tetrad analysis of the strains that are heterozygous for the alleles of *PUF5*, *CLB2*, *FKH1*, and *LRG1* (A), *PUF5*, *CLB2*, *FKH2*, and *LRG1* (B), *PUF5*, *CLB2*, *HFI1*, and *LRG1* (C), *PUF5*, *CLB2*, and *HIR1* (D), *PUF5*, *CLB2*, *STE12*, and *LRG1* (E). The cells were sporulated, dissected on a YPD plate containing 10% sorbitol, and cultured at 30°C for 3 days. (TIFF)

**S3 Fig. *IXR1* functions independently of *LRG1*, a target of Puf5.** The effect of the *ixr1*Δ mutation and the *lrg1*Δ mutation on cell growth at 37°C. The sets of the strains obtained from the tetrad analysis shown in Fig 5B were picked on a YPD plate containing 10% sorbitol and incubated at 25°C for 1 day. Then, the plate was replicated to a YPD plate and incubated at 37°C for 1 day. (TIFF)

**S4 Fig. Genetic interaction between *IXR1* and *FKH1/FKH2*.** The tetrad analysis of the strains that are heterozygous for the alleles of *IXR1*, *FKH1*, and *FKH2*. The cells were sporulated, dissected on a YPD plate containing 10% sorbitol, and cultured at 30°C for 3 days. (TIFF)

**S5 Fig. Western blot images of Puf5 and Ixr1 during the cell cycle.** (A) The Puf5 protein level in the synchronized *bar1*Δ cell integrated *PUF5*-Myc-*ADH1* 3′ UTR gene. No tag sample was loaded as a negative control. The arrowheads show two different bands of Puf5-Myc protein observed. Band 1 and band 2 were quantified and normalized with the Pgk1 protein level. Fold change is presented in Fig 12B. (B, C) The Ixr1 protein level in the synchronized *bar1*Δ cell (B) and *bar1*Δ *puf5*Δ mutant (C) harboring the YCplac33-*IXR1*-HA-*IXR1* 3′ UTR plasmid. No tag sample was loaded as a negative control. The arrowheads show three different bands of Ixr1-HA protein observed. Band 1 and band 2 were quantified and normalized with the Pgk1

protein level. Fold change is presented in Fig 13C. (D, E) The comparison of the Ixr1 protein level (D) and quantitative analysis data of Ixr1-HA protein (E) between the synchronized *bar1Δ* cell and *bar1Δ puf5Δ* mutant harboring the YCplac33-*IXR1*-HA-*IXR1* 3′ UTR plasmid. 100 min samples of each strain were loaded. The arrowheads show three different bands of Ixr1-HA protein observed. Band 1 and band 2 were quantified and normalized with the Pgk1 protein level. The data shows the fold change of Ixr1-HA protein relative to the protein level in the *bar1Δ* cell (E).
(TIFF)

## Acknowledgments

We thank all the members of the Molecular Cell Biology Laboratory for valuable discussions.

## Author Contributions

**Conceptualization:** Megumi Sato, Kenji Irie.

**Data curation:** Megumi Sato, Kenji Irie.

**Formal analysis:** Megumi Sato, Kenji Irie.

**Funding acquisition:** Kenji Irie.

**Investigation:** Megumi Sato, Kaoru Irie, Kenji Irie.

**Project administration:** Kenji Irie.

**Supervision:** Kenji Irie.

**Validation:** Megumi Sato.

**Visualization:** Megumi Sato.

**Writing – original draft:** Megumi Sato, Kenji Irie.

**Writing – review & editing:** Megumi Sato, Yasuyuki Suda, Tomoaki Mizuno, Kenji Irie.

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
