## [Decision Letter · Decision Letter 0]

22 Apr 2022

Dear Dr Irie,

Thank you very much for submitting your Research Article entitled 'The RNA-binding protein Puf5 and the HMGB protein Ixr1 contribute to cell cycle progression through the regulation of cell cycle-specific expression of CLB1 in Saccharomyces cerevisiae' to PLOS Genetics.

The manuscript was fully evaluated at the editorial level and by independent peer reviewers. The reviewers appreciated the attention to an important problem, but raised some substantial concerns about the current manuscript. Based on the reviews, we will not be able to accept this version of the manuscript, but we would be willing to review a much-revised version. We cannot, of course, promise publication at that time.

If you decide to revise the manuscript for further consideration at PLOS Genetics, please aim to resubmit within the next 60 days, unless it will take extra time to address the concerns of the reviewers, in which case we would appreciate an expected resubmission date by email to plosgenetics@plos.org.

[LINK]

We are sorry that we cannot be more positive about your manuscript at this stage. Please do not hesitate to contact us if you have any concerns or questions.

Yours sincerely,

Aaron P. Mitchell, PhD

Associate Editor

PLOS Genetics

Gregory P. Copenhaver

Editor-in-Chief

PLOS Genetics

Reviewer's Responses to Questions

**Comments to the Authors:**

Reviewer #1: This article by Sato et. al. dissects the role of Puf5, a pumilio/FBF domain-containing RNA-binding protein, in cell cycle progression in Saccharomyces cerevisiae. The authors utilize a series of genetic approaches to demonstrate that overexpression of CLB2 suppresses the puf5∆ phenotype at the level of promoter activation. Furthermore, they identified that IXR1 deletion rescues the puf5∆clb2∆ double mutant phenotypes, and demonstrate that Puf4 regulates IXR1 post-transcriptionally. Overall, the results presented in this manuscript seem compelling and well organized. Below are minor suggestions and corrections; otherwise, the manuscript is technically sound, the conclusions are supported by the data, and the interplay between an RBP and a transcriptional suppressor is interesting and novel.

- Because authors use the ADH1 3’ UTR as a control, it would be informative to show the expression of ADH1 in all different mutants as a supplemental figure to show that its regulation is not altered in different mutants.

- The authors cite reference #7 for the interaction between IXR1 and Puf5. Does Puf5 also bind to any CLB or CLN mRNAs?

- The authors show that the cell-cycle-dependent expression of some genes is dependent on Puf5 and Ixr1. Does the expression of PUF5 mRNA or Puf5 protein change throughout the cell cycle?

- Line 48: RNA binding proteins bind to both 5’ and 3’ UTRs, not only 3’ UTRs.

- The authors should explain the genomic DNA library preparation and how they performed the screening in the methods section.

- Line 211: “CLB1 suppressed the temperature sensitivity phenotype”. This seems to be only true at 35˚C and not 37˚C. Authors could comment more on this and maybe mention it in the discussion.

- Line 289: Typo, should read as “promoter”.

- Typo, Figure 7D y-axis label should read as “Fold”.

- “expressions” is used several times interchangeably with “expression”. It should always be singular.

- Line 347: Fig 7C does not show the correct information. This should refer to Fig 6D.

- Overall, authors should revise their discussion to reflect how their work fits into the field of RNA binding proteins and cell cycle regulation.

Reviewer #2: Sato et al report on a novel cell cycle regulation mediated by the RNA binding protein Puf5. Overall, I found the manuscript interesting and the model is in general supported by the data. However, some key experiments to directly connect the different parts of the pathway are missing (see below). In addition, the authors need to provide more information (data and background) to allow easier readability of the manuscript.

Major points:

The screening procedure is explained very minimally, e.g. how was the screening library constructed (or cite relevant paper). Why was Lrg1 co-transformed with the screening library? For a better understanding, it would be good to add a description in the results section and to expand the M&M with details of the library. Were SSD1, CLB2 and ZDS1 the only hits in the screen. It would be nice to include a table as part of the supplements with all the hits (even though they might not be validated) for future reference.

Figure 4 examines a potential role in transcriptional regulation of Clb1 mRNA. As PUF5 has been implicated to impact mRNA stability, this result in itself is not convincing enough to rule out any post-transcriptional roles of PUF5 in this experiment. The authors should provide additional controls, e.g. using the ADH1 promoter in the same fashion.

To identify potential regulators of Clb1, the authors screened SGD and selected 16 potential regulators (Fig. 5A) and claim that IXR1, FKH1 and FKH2 increase expression in a puf5D mutant, but do not show the corresponding data. This must be included in the manuscript (for all regulators tested). Same holds true for the subsequent tetrad analysis of the fkh1/2 deletion (line 316/317).

Fig. 5E should have the full set of controls, including pu5D, clb2D and puf5 clb2D cells to directly compare the phenotypes.

I think the experiment using 3’UTR swaps to analyse the PUF5-dependency on Ixr1 expression is very convincing. To show more directly that Ixr1 is indeed a target of Puf5, the authors should consider to perform a Puf5 RIP and analyse enrichments of Ixr1 mRNA. More important would be to directly show that Ixr1 regulates Clb1 transcription and that Ixr1 is indeed recruited to Clb1 promoter (with increased levels in puf5D).

Minor points:

Tetrad dissection was performed at 30oC according the figure legend. However, in the text, it is written in a way that suggests 25oC (line 217), which renders Fig. 1D a bit confusing.

As the effect of puf5D appears to be stronger on the level of protein compared to RNA, it would be good to include a western blot of Clb1-HA expression upon over expression of PUF5.

Clb1 over expression rescues puf5D clb2D growth defect, which is also manifested in elongated shaped cells. Can ZSD1 or SSD1 over expression (which perturb similar polarity pathways) also rescue the growth defect of the double knock out?

The authors test whether deletion of Lrg1 might have an effect on the genetic interaction between puf5 and clb2 - it is not clear to me why this has been included. The corresponding section in the text should be extended (line 280ff)

To better compare the expression changes in the alpha-factor release experiments, I would suggest to combine figure 8,9 and 10, especially as figure 10 serves as a control for the results in Figure 9.

Reviewer #3: The review report is uploaded as an attachment file.

**Have all data underlying the figures and results presented in the manuscript been provided?**

Reviewer #1: **No: **I did not find any spreadsheets containing data that are represented in bar or line graphs within the manuscript.

Reviewer #2: **No: **I have not found underlying data for all barplots presented in this paper.

Reviewer #3: Yes

PLOS authors have the option to publish the peer review history of their article (what does this mean?). If published, this will include your full peer review and any attached files.

Reviewer #1: No

Reviewer #2: No

Reviewer #3: No

***Please note - two reviewers mentioned that the underlying data for your bar graphs was not included as a supplemental file.  Please attend to that if/when you submit a revision.

---

## [Decision Letter · Decision Letter 1]

11 Jul 2022

Dear Dr Irie,

Thank you very much for submitting your Research Article entitled 'The RNA-binding protein Puf5 and the HMGB protein Ixr1 contribute to cell cycle progression through the regulation of cell cycle-specific expression of CLB1 in Saccharomyces cerevisiae' to PLOS Genetics.  The reviewers were satisfied with the revision for the most part, but please see requests below from Rev 3 regarding minor issues of interpretation and phrasing.  The issue for the FKH genes is that they may act downstream of Ixr1 or potentially in parallel.  Please qualify your statements suitably.  The other issues seem very straightforward.  Feel free to contact me (APM) if you have any questions.  I do not plan to send the revision back to the reviewers but just read it myself.

[LINK]

Yours sincerely,

Aaron P. Mitchell, PhD

Associate Editor

PLOS Genetics

Gregory P. Copenhaver

Editor-in-Chief

PLOS Genetics

Reviewer's Responses to Questions

**Comments to the Authors:**

Reviewer #1: The authors have addressed all concerns in this revision.

Reviewer #2: The authors have adequately addressed my previous concerns and I support publication.

Reviewer #3: The authors applied almost all revision, especially addition of data, suggested by the reviewer to clear the major and minor comments. However, the revision caused several parts to be mended.

1) Although the authors excluded Fkh1 and Fkh2, transcription regulators of CLB1 mRNA, from their primary analyses in p. 17, they picked up these factors again p. 19, especially pointing that parallel contribution of the two genes from the fact that synthetic phenotype of the double deletion. Such development of the point of argument confuses readers.

Furthermore, from p. 19, line 464 to p. 20, line 466, they tried to place Fkh1 and Fkh2 downstream of Ixr1 merely from the fact that IXR1-deletion did not suppress the fkh1 fkh2 double mutant. They failed to appreciate another possibility that Fkh1 and Fkh2 are unrelated to the Ixr1-axis of CLB1 transcriptional regulation.

2) The authors used a word "invariable" to describe expression patterns of IXR1 mRNA and Ixr1-HA protein. However, significant but cell-cycle independent fluctuation of the mRNA amounts is obvious in Fig. 13. They must choose words more precisely.

3) The unit of temperature, centigrade (°C), did not appear properly in many parts of the text; for example, p. 8, line 166.

4) The mark for "less than or equal to" (≤ ) did not appear properly in p. 13, lines 291 and 293.

**Have all data underlying the figures and results presented in the manuscript been provided?**

Reviewer #1: Yes

Reviewer #2: Yes

Reviewer #3: Yes

PLOS authors have the option to publish the peer review history of their article (what does this mean?). If published, this will include your full peer review and any attached files.

Reviewer #1: **Yes: **John C. Panepinto

Reviewer #2: No

Reviewer #3: No

---

## [Editor Report · Decision Letter 2]

14 Jul 2022

Dear Dr Irie,

We are pleased to inform you that your manuscript entitled "The RNA-binding protein Puf5 and the HMGB protein Ixr1 contribute to cell cycle progression through the regulation of cell cycle-specific expression of CLB1 in Saccharomyces cerevisiae" has been editorially accepted for publication in PLOS Genetics. Congratulations!

Yours sincerely,

Aaron P. Mitchell, PhD

Associate Editor

PLOS Genetics

Gregory P. Copenhaver

Editor-in-Chief

PLOS Genetics

Comments from the reviewers (if applicable):

**Data Deposition**

http://datadryad.org/submit?journalID=pgenetics&manu=PGENETICS-D-22-00414R2

**Press Queries**

---

## [Editor Report · Acceptance letter]

24 Jul 2022

PGENETICS-D-22-00414R2 

The RNA-binding protein Puf5 and the HMGB protein Ixr1 contribute to cell cycle progression through the regulation of cell cycle-specific expression of CLB1 in Saccharomyces cerevisiae 

Dear Dr Irie, 

We are pleased to inform you that your manuscript entitled "The RNA-binding protein Puf5 and the HMGB protein Ixr1 contribute to cell cycle progression through the regulation of cell cycle-specific expression of CLB1 in Saccharomyces cerevisiae" has been formally accepted for publication in PLOS Genetics! Your manuscript is now with our production department and you will be notified of the publication date in due course.

With kind regards,

Zsofia Freund

PLOS Genetics

On behalf of:
